



# VAHCOLI, a new concept for lidars: technical setup, science applications, and first measurements

Franz-Josef Lübken[1] and Josef Höffner[1]

[1]Leibniz-Institute of Atmospheric Physics, Schloss-Str. 6, Kühlungsborn (URL: www.iap-kborn.de), Germany

**Correspondence:** Franz-Josef Lübken (luebken@iap-kborn.de)

**Abstract.** A new concept for a cluster of compact lidar systems named VAHCOLI (Vertical And Horizontal COverage by LIdars) is presented which allows to measure temperatures, winds, and aerosols in the middle atmosphere (~10-110 km) with high temporal and vertical resolution of minutes and some tens of meters, respectively, simultaneously covering horizontal scales from few hundred meters to several hundred kilometers ('four-dimensional coverage'). The individual lidars ('units') being used in VAHCOLI are based on a diode-pumped alexandrite laser currently designed to detect potassium ($\lambda$=770 nm), as well as on sophisticated laser spectroscopy measuring all relevant frequencies (seeder laser, power laser, backscattered light) with high temporal resolution (2 ms) and high spectral resolution applying Doppler-free spectroscopy. The frequency of the lasers and the narrow-band filter in the receiving system are stabilized to typically 10–100 kHz which is a factor of roughly $10^{-5}$ smaller than the Doppler-broadened Rayleigh signal. Narrow-band filtering allows to measure Rayleigh and/or resonance scattering separately from the aerosol (Mie) signal, all during night and day. Lidars used for VAHCOLI are compact (volume: ~1 m$^3$) and are multi-purpose systems employing contemporary electronical, optical, and mechanical components. The units are designed to autonomously operate under harsh field conditions at remote locations. An error analysis with parameters of the anticipated system demonstrates that temperatures and line-of-sight winds can be measured from the lower stratosphere to the upper mesosphere with an accuracy of $\pm$(0.1–5) K and $\pm$(0.1–10) m/s, respectively, increasing with altitude. We demonstrate that some crucial dynamical processes in the middle atmosphere, such as gravity waves and stratified turbulence, can be covered by VAHCOLI with sufficient temporal/vertical/horizontal sampling and resolution. The four-dimensional capabilities of VAHCOLI allow to perform time-dependent analysis of the flow field, for example employing Helmholtz decomposition, and to carry out statistical tests regarding intermittency, helicity etc. First test measurements under field conditions with a prototype lidar being built for VAHCOLI were performed in January 2020. The lidar operated successfully during a six week period (night and day) without any adjustment. These observations covered a height range of ~5–100 km and demonstrate the capability and applicability of this unit for the VAHCOLI concept.

## 1 Introduction

Lidars (light detection and ranging) have been applied in atmospheric research since many years. Here we concentrate on the middle atmosphere, namely on the altitude range 10–120 km. Different techniques have been used to measure, e. g., tempera-



tures, winds, aerosols, metal densities, as well as atmospheric characteristics deduced from prime observations, such as grav-
ity waves or trends (see, for example, Hauchecorne and Chanin, 1980; von Zahn et al., 1988; She et al., 1995; Keckhut et al.,
1995; Gardner et al., 2001; Collins et al., 2009, and references therein). Backscattering from molecules (Rayleigh, Raman),
from aerosols (Mie), as well as resonance scattering from metal atoms in the upper mesosphere/lower thermosphere have been
applied to deduce number densities (background and metals), temperatures, winds, and important characteristics of aerosols

(size, number densities) such as noctilucent clouds (NLC) and polar stratospheric clouds (PSC). In the standard setup lidars
perform measurements in the vertical but oblique soundings have also been applied occasionally, e. g. for the lidars at ALO-
MAR (Arctic Lidar Observatory for Middle Atmosphere Research) and at the Starfire Optical Range (von Zahn et al., 2000;
Chu et al., 2005). Typical altitude and time intervals are 100 m and 10 min, respectively, where much better altitude/time
resolutions have occasionally been achieved. In summary, lidars measure highly relevant atmospheric parameters with high

temporal and vertical resolution. The main disadvantage of lidars is that observations are normally made in a single column
with very limited horizontal coverage, and often only during darkness and, of course, only during clear sky conditions. Lidars
have also been developed for applications on airplanes and balloons which can travel substantial horizontal distances but are
limited in resolving temporal/spatial ambiguities (see, for example, Shepherd et al., 1994; Voigt et al., 2018; Fritts et al., 2020).
Furthermore, these applications are rather complex and costly and are therefore performed only sporadically. Lidars on satel-

40 lites, e. g., CALIPSO (Cloud-Aerosol Lidar and Infrared Pathfinder Satellite Observations) have also been developed and have
been used to measure, e. g., PSC (see overviews in Weitkamp, 2009; Winker et al., 2009). However, so far the application of
these satellite lidars regarding middle atmosphere research is restricted due to their limited height coverage. For example, the
spaceborne wind lidar mission ADM-Aeolus (Atmospheric Dynamics Mission Aeolus) aims at observing winds up to 30 km
(Reitebuch, 2012). Some ground-based techniques have been developed to cover larger horizontal distances in the middle at-

45 mosphere, like the Advanced Mesospheric Temperature Mapper (AMTM) based on airglow emissions in the mesopause region
(Pautet et al., 2015). Multistatic radars are now available to measure winds in the upper mesosphere/lower thermosphere with
extended horizontal coverage (Chau et al., 2017; Vierinen et al., 2019). Compared to lidars these techniques cover a rather lim-
ited height range. Quasi-permanent wind observations in the stratosphere and mesosphere are performed applying microwave
technology, however, with rather poor temporal and height resolution of hours to days and several kilometers, respectively (see,

e.g., Rüfenacht et al., 2018, and references therein).

   The VAHCOLI concept of placing a cluster of lidars ('units') at various locations relies crucially on individual instruments
being specially designed and developed for this purpose. We therefore first describe the technical concept and performance
of such a unit. The main idea is to use modern technology to drastically miniaturize and simplify all lidar sub-components
such as laser, telescope, and receiver system, whereas the measurement capabilities shall be similar, or even better compared to

55 contemporary existing lidars. Furthermore, sophisticated laser spectroscopy methods shall be applied to measure the spectrum
of the backscattered signal with high spectral resolution, accuracy, and sampling rate. This allows, for example, to separate
backscattering from molecules (Rayleigh) and aerosols (Mie) and to measure Doppler widths and shifts of the backscattered
signal simultaneously. A VAHCOLI-unit shall be robust so that it can easily be operated under field conditions with minimal
infrastructure in automatic day and night operation. A network of several of these lidars shall be placed at various locations,



and each lidar shall be employed with several oblique beams, so that (apart from vertical) a substantial horizontal range is covered simultaneously. Finally, the lidar must be cost-effective and shall operate for long periods of time (several months or longer) without maintenance. It must still be able to measure various scattering mechanisms to monitor the atmosphere from approximately 10 to 100 km.

## 2    Selected technical features of a VAHCOLI-unit

### 2.1    General

A compact setup for a VAHCOLI-unit is chosen with an adequate choice of optical, electronical and mechanical components. If available, off-the-shelf components regarding optics, mechanics, electronics etc. are used. Major parts of the mechanics and housing are produced by 3D printing. This allows cost effective and flexible modifications of mechanical components and mountings of optical subsystems if, for example, the application has to be adjusted to certain scientific requirements or to

70 specific background conditions. The overall goal is to build a general purpose lidar to allow for observations applying Rayleigh, Mie, and resonance scattering. A first prototype of a VAHCOLI-unit is now available and has recently been tested under field conditions (see section 3). A photo of the prototype and a technical drawing are shown in Fig. 1.

Fig. 1

#### 2.1.1    Spectral characteristics of scattered signals and lidar components

The lidars being used for the VAHCOLI concept rely on a careful consideration and measurement of various spectral charac-

75 teristics of laser frequencies, spectral filters, and backscattered light. We therefore briefly recollect the spectral features of the main scattering processes and instrumental components involved (see Fig. 2). The spectral line width (FWHM = full width

Fig. 2

of half maximum) of the Doppler-broadened line due to scattering on molecules (Rayleigh) is typically $\Delta\nu_m \sim$1500 MHz (for $\lambda$=770 nm, the potassium resonance wavelength currently being used) which is given by the Maxwell-Boltzmann velocity distribution. The line width is proportional to $\sqrt{\frac{T}{m_m}}$, i.e., it can be used to measure temperatures ($T$ = temperature ; $m_m$ =

mean mass of an air molecule). The line widths of resonance lines of metal atoms are again given by Doppler-broadening on top of broadening due to atomic physics processes such as natural lifetime and hyperfine-structure. For potassium, the FWHM is appr. 1000 MHz (von Zahn and Höffner, 1996). The line-width of scattered light caused by aerosols is much smaller since these aerosols are much heavier compared to molecules. Stratospheric aerosol particles have radii on the order of $0.1\mu$m which corresponds to a mass of roughly $7{\cdot}10^{-6}$ ng. This is a factor of $\sim$1.5$\cdot$10$^8$ larger compared to the mass of an air molecule.

Therefore, the spectral width of the aerosol signal $\Delta\nu_a$ is roughly $\Delta\nu_a = \Delta\nu_m \cdot \sqrt{\frac{m_a}{m_m}} \sim$ 100 kHz for $\Delta\nu_m$=1500 MHz. The Doppler shift, $d\nu$, of the backscattered signal due to background winds is $d\nu= 2 \cdot \nu_o \cdot v/c$ which allows to measure line-of-sight winds ($\nu_o$ = laser frequency, $v$ = background wind). For a laser wavelengths of $\lambda_o$=770 nm ($\nu_o$=389.28 THz) and applying Rayleigh or Mie scattering, this shift is $d\nu$= 2.6 MHz for a wind of $v$=1 m/s. We also need to take into account the spectral widths of the instrumental components (see later), namely the spectral widths of the diode-pumped alexandrite laser

($\sim$3.3 MHz) and the high resolution spectral filter ('confocal etalon') in the receiver system (appr. 7.5 MHz). For comparison,





the Fourier-transform width of a laser pulse with a length of 1000 m (100 m, 1 m) is roughly 60 kHz (600 kHz, 60 MHz). According to their spectral characteristics, the scattering mechanisms mentioned above are used to measure temperatures and winds from Rayleigh scattering ('Doppler-Rayleigh'), temperatures and winds (plus metal number densities) from resonance scattering ('Doppler-resonance'), and winds (plus aerosol densities) from Mie scattering ('Doppler-Mie').

### 2.1.2 General lidar setup

A key idea behind the lidars being used for VAHCOLI is that all relevant frequencies and spectra (e. g., seeder laser, power laser, backscattered light from the atmosphere, narrowband filter, reference spectrum) are controlled and measured with high precision (see later). The atmospheric signal and laser parameters are measured (or actively controlled for the latter) for every single laser pulse. This allows, for example, to measure the widths and shifts of the spectrally broadened lines and, in parallel, determine the spectral characteristics of the receiver in the time between two laser pulses. Narrow band spectral filtering and a small field-of-view is applied to reduce the background, which allows to use high repetition frequencies and comparatively low power laser energies. The laser can be tuned to any given frequency within a large frequency range, and spectra are observed at these mean frequencies with great detail. This provides, for example, the opportunity to measure the signals from Rayleigh and Mie scattering separately (see section 3).

The frequency of the power laser is controlled by a so-called seeder laser. The seeder laser is used i) to control the frequency of the power laser, and ii) to measure the spectral specifications of the entire optical path in the receiver system (including the filter characteristics of the etalons involved) immediately before these filters are being used to measure the spectrally broadened and shifted backscattered signal from the atmosphere (Rayleigh, Mie, or resonance).

The seeder laser, the power laser, and the signal from the atmosphere are fed into a receiver system (see Fig. 3) which consists of various optical components such as an interference filter (to block a large part of the solar background spectrum not wanted), a broad band solid etalon with a FWHM close to the Doppler width of the atmospheric molecule line, and a narrow band confocal etalon (FWHM $\sim$7.5 MHz). The largest part of the incoming light is reflected by the confocal etalon and creates a signal at the detector $D_{R-R}$, whereas a small part is transmitted and is measured by the detector $D_{Mie}$ (see Fig. 3). The frequency of the seeder laser is tuned up and then tuned down again. The amplitude of this tuning is, for example, 2000 MHz in order to cover the Doppler broadened Rayleigh signal. The seeder is used to control the frequency of the power laser for every single pulse, i. e., every 2 ms. The spectral characteristics of the seeder laser, e. g., the frequency as a function of time during tuning up and down, are known precisely due to comparison with a high-resolution Doppler-free polarization spectrum of potassium, which also serves as an absolute frequency reference in case of resonance scattering for potassium. The parameters controlling the seeder frequencies can easily be adjusted by software according to the scientific requirements. The lidar parameters, e. g., the range of the frequency scan, can be optimized for the measurement of Doppler winds on aerosols, or for resonance temperatures, or for a simultaneous observation of Rayleigh, Mie, and resonance scattering.

The frequency of the power laser is nearly identical to the frequency of the seeder laser because a novel cavity control technology called Advanced Ramp and Fire (ARF) is applied. A precursor version of this technology was used since 1998 for the flashlamp-pumped alexandrite ringlaser of the IAP, and later also for Nd:YAG (Nicklaus et al., 2007). ARF allows to

Fig. 3





control the mean frequency shift between seeder and power laser, and it is not limited to a single laser frequency. This allows
a fast tuning of the power laser to a wide range of frequencies from one pulse to the other. We currently use a maximum
tuning rate of 1000 MHz per milli-second, which can be increased if required. An example of controlling and measuring the
power laser frequency is shown in Fig. 4. The seeder laser (and thereby the power laser) is tuned across the confocal etalon | Fig. 4 |
where the frequency range of the tuning (=100 MHz) was chosen such that it covers the spectral width of the confocal etalon.
The frequency sampling covers frequencies with a difference of 2 MHz. As can be seen in Fig. 4 the advanced ramp-and-fire
technique ensures that the frequencies of the power laser pulses are indeed very close to the nominal frequencies, $\nu_i$ (within
less than 100 kHz), namely within the width of each red line in Fig. 4. The intensity distribution as a function of $\nu_i$ is given by
the convolution of the spectral width of the confocal etalon and the spectral width of the power laser (the spectral width of the
seeder laser is only $\sim$100 kHz and can be ignored in this context). Fig. 4 demonstrates that the frequency control of the power
laser by the seeder laser works very successfully, namely within less than approximately 100 kHz.

The temporal stability of the laser frequency control is demonstrated in Fig. 5 where measurements of the frequencies of the | Fig. 5 |
seeder laser and the confocal etalon are shown. More precisely, in the upper panel the difference between the nominal seeder
laser frequency and the actual true frequency is shown, where the latter is determined by comparison with high precision
Doppler-free spectroscopy. Data points are shown for $\sim$3 min with a temporal resolution of 1/10 s. The mean offset between
nominal and true frequencies is only 21.44 kHz with a RMS variation of 170 kHz. The same procedure is repeated for the
confocal etalon (lower panel in Fig. 5), i. e., the seeder laser is directed into the confocal etalon, and nominal and 'true'
frequencies (from the seeder laser) are compared to each other. The mean uncertainty of the confocal etalon frequency is
187.92 kHz which corresponds to an uncertainty in wind measurements of 0.072 m/s. In fact, the final contribution of this
uncertainty to the wind error is even smaller since the offset (in this case 187.92 kHz) is measured and later considered in the
data reduction procedure.

Finally, the spectrum received from the atmosphere is compared with the spectral characteristic of the instrument including
laser line width, spectral filters etc. For Rayleigh and Mie scattering it is not necessary to know the absolute frequency of
the laser light and the absolute frequency position of the etalon's transmission functions. It is only important to know the
frequency of the pulsed laser relative to the frequency of the spectral filters which is achieved by the procedure described
above. On the other hand, resonance scattering requires frequency measurements on an absolute frequency scale which is
achieved by applying Doppler-free polarization spectroscopy to an atomic absorption line of potassium, which in turn is used
to control the output of the seeder laser and the spectral filters with an accuracy of a few kHz (see above).

## 2.2   Laser specifications

A VAHCOLI-unit requires a compact, efficient, and high performance laser designed for atmospheric applications. For Doppler-
Mie the laser should preferentially have an exceptionally small line-width. For Doppler-resonance the laser must be tunable to
an atomic absorption line. We have developed a highly efficient, narrow-band diode-pumped alexandrite ring laser in coopera-
tion with the Fraunhofer Institut for Laser Technology in Aachen (Höffner et al., 2018; Strotkamp et al., 2019). The laser head
includes various subsystems such as Q-switch driver, cavity-control, power measurement, and a beam expansion telescope all





of which are placed in a sealed housing for touch free operation over long periods. The laser head is pumped via a fiber cable
connected to a separate diode array acting as an optical pump. The beam profile of the laser is nearly perfect with very little
aberration ($M^2$=1.1). The spectral width is ~3.3 MHz with a pulse length of ~780 ns. In Q-switch mode, the variation of the
output power from pulse to pulse is only 0.2%. A first test of the robustness of the laser was performed when it was transported
from Aachen to Kühlungsborn in early 2020. After a transport of nearly 700 km in a standard truck the laser performed without
any degradation (see section 3). Thereafter, the entire lidar was aligned and operated successfully during a six week period
without any further adjustment.

## 2.3  Telescope and receiver

The field-of-view (fov) of the telescope is currently 33 $\mu$rad which corresponds to a diameter of 3.3 m at a distance of 100 km.
The accuracy to keep the laser beam inside the field of view of the telescope is better than 10 cm at 100 km which corresponds
to a position accuracy of better than 1 $\mu$rad. This is achieved as follows: the photons being scattered by 180 degree from the
atmosphere follow the optical path of the outgoing laser-beam but in retrograd direction. The light from the atmosphere is
separated from the outgoing laser pulse using its polarization characteristics. This implies that outgoing and incoming light
paths are automatically co-aligned and no active control of the outgoing laser beam on a pulse-to-pulse basis is needed. Slow
drifts of the laser beam relative to the optical axis of the telescope caused by, for example, temperature drifts are compensated
for by a control loop (maximizing the atmospheric signal) with a time constant of few minutes. The current plan is to have 5
telescopes with different viewing directions which are fed subsequently (switching within 1 ms) by one laser. Only one receiver
system (and one transmitter) will be needed.

The prime mirrow (diameter = 50 cm) and related optics are integrated into a system which is manufactured by large scale
3D printing. Complex thermal balancing considerations ensure that the telescope (optics and walls) are stabilized to the outside
temperature by maintaining an active airflow through the cube. This prevents convection and turbulence and also keeps dust,
snow, and sea salt away from the optics. On the other hand, the mechanical structures supporting the primary and secondary
mirrors are stabilized to the temperature inside the main housing. This eliminates the need to realign the telescope when
ambient temperatures are changing, for example, from day to night. In summary, the mechanical design and thermal balancing
allow to operate the lidar under harsh conditions at a wide range of ambient temperatures during day and night.

The most important components of the receiver are the spectral filters in combination with other optical systems such as the
seeder laser and the Doppler free spectroscopy (see Fig. 3). All components fit into a compact, optically tight, dust free, and
lightweight housing of 15 x 15 x 80 cm which is manufactured by 3D printing together with all mechanical mounts for the
optical components (~75 in total). Avalanche Photodiodes (APD) are used for counting photons.

## 2.4  Data acquisition and lidar control

After a laser pulse has been released, it takes 1 ms until photons scattered at a distance of 150 km arrive at the detector. This
implies that the maximum possible pulse repetition frequency is $k^{max}$=1000 laser pulses per second (we use 500 per second),
assuming that only one laser pulse is in the air at any given time. Photons from a single pulse scattered from a height range $\delta z$





arrive at the detector in a time interval of $\Delta t = 2 \cdot \delta z / c$. During arrival the number of photons $N_{ph}$ from the height range $\delta z$ create a count rate $R_{sgl}$ at the detector of $R_{sgl} = N_{ph} / \Delta t = N_{ph} / (2 \cdot \delta z / c)$. Here and in the following we ignore any impact by the dead-time of the detector. The time between two pulses is given by dt=1/k where k is the pulse repetition frequency. Therefore

the effective number of photons counted per time interval is reduced relative to $R_{sgl}$ by a factor of $\Delta t / dt = (2 \cdot \delta z / c) \cdot k$. For example, for $\delta z$=200 m and k=500/s the reduction factor is 1/1500. In other words, the number of photons, $N_{ph}$, counted per integration time $\delta t$ and height interval $\delta z$ is related to the count rate $(R)$ and the pulse repetition rate $(k)$ via

$$N_{ph} = R \cdot \frac{\Delta t}{dt} \cdot \delta t = R \cdot (2 \cdot \frac{\delta z}{c} \cdot k) \cdot \delta t \qquad (1)$$

For technical reasons the count rate of typical detectors (PMT, APD) is limited to approximately $R^{max}$=10$^7$ Hz. As

mentioned before, the maximum pulse repetition rate is given by the uppermost altitude $(z_{max})$ wanted, $k^{max}$=1000/s for $z_{max}$=150 km. Higher pulse repetition frequencies may be chosen if the maximum altitude is reduced. According to equation 1 this leads to a larger number of photons at a given altitude. The receiver relies on high speed single photon data acquisition system with compression for fast analysis. Each pulse is stored with 1 m altitude resolution and with further information regarding, for example, pulse energy as well as frequency and FWHM of the laser pulse. Each VAHCOLI-unit is connected to

the internet and, if necessary, can be controlled and operated in real time. This includes the frequency control of the seeder and power lasers as well as the filter system (see above). The entire receiver system (actually the entire lidar) is based on a single standard PC with integrated commercially available electronics. Data from the lidar are automatically downloaded to a remote server.

## 2.5 Measuring principle

### 2.5.1 Temperatures and winds from Rayleigh and resonance scattering

A standard method to derive temperature profiles from measured altitude profiles of number densities is based on the (downward) integration of the hydrostatic equilibrium equation. Since only relative number densities are relevant here, the lidar count rates from Rayleigh scattering can be applied, after taking into account the square of the distance (lidar equation). Some uncertainties are introduced by the unknown temperature at the top of the profile (also called 'start temperature') which, however,

decrease exponentially with altitude. A more detailed error analysis for Rayleigh temperatures is presented in section 4.2. Note that narrow spectral filtering allows to separate the Rayleigh signal from the Mie signal (see below). This implies that Rayleigh temperatures are derived even in the presence of aerosols. Since the spectral width of the Rayleigh signal is proportional to $\sqrt{T}$ it can also be used to measure temperatures. This is planned for the future, together with a comparison of temperatures from integration (hydrostatic equation).

Winds are measured by lidars by detecting the spectral shift of the backscattered light (Rayleigh or resonance or Mie). This is rather challenging since this shift is normally small compared to the spectral width of the backscattered signal. Various techniques have been developed to measure the spectral shift, e. g. employing double-edge or single-edge or vapor filters (see,





for example, Chanin et al., 1989; She and Yu, 1994; Baumgarten, 2010). In our first measurements presented in section 3 we concentrated on measuring winds by detecting the spectral shift of the narrow band aerosol signal.

Resonance scattering on metal atoms (K, Fe, Na) has frequently been applied to derive number densities and temperatures in the altitude range of roughly 80 to 120 km by measuring the Doppler width of the backscattered light (see, for example, Fricke and von Zahn, 1985; von Zahn et al., 1988; Alpers et al., 1990; She et al., 1990; Clemesha, 1995; Chu et al., 2011; Höffner and Lautenbach, 2009). Different from these lidars, VAHCOLI can observe winds and temperatures from resonance scattering in the presence of aerosols, namely NLC. The resonance scattering application of VAHCOLI is based on our experi-
ence with a potassium lidar being operated at several locations, for example on the research vessel Polarstern or in Spitsbergen (Höffner and von Zahn, 1995; von Zahn and Höffner, 1996; Lübken et al., 2004; Höffner and Lübken, 2007). The technique has been improved substantially for a VAHCOLI-unit by applying high temporal and high spectral resolution detection of Doppler broadening (temperatures) and Doppler shift (line-of-sight winds). See section 2.1.1 for more details.

### 2.5.2   Aerosol parameters and winds

VAHCOLI is designed to also measure the presence of aerosols, more precisely background aerosols, polar stratospheric clouds (PSC), and noctilucent clouds (NLC). Precise and fast measurements of the spectrum of the filters allows to position the narrow spectral filter (few MHz) exactly at the position of the Mie peak related to aerosol backscattering, as is shown in Fig. 2. Since the Mie spectrum in the stratosphere is also very narrow (typically 0.1 MHz, see above) only the backscattered signal from aerosols is detected, whereas nearly all of the Rayleigh scattering is blocked (this implies that the solar background signal is
negligible which is known as 'solar blind'). Therefore, Mie scattering is detected irrespective of the Rayleigh signal (and vice versa). The precise measurement of spectra allows to derive line-of-sight winds from the Doppler shift of the Mie peak (see section 3). In the future, we envisage multi-color observations of PSC and NLC to deduce particle characteristics such as size and number densities (see, for example, von Cossart et al., 1999; Alpers et al., 2000; Baumgarten et al., 2010).

### 2.5.3   Metal densities

Resonance scattering on metal atoms in the upper mesosphere/lower thermosphere is applied to derive metal number density profiles. We have used this technique mainly to observe potassium ($\lambda$=770 nm) and iron ($\lambda$=386 nm), but other metals have also been measured (see, e.g., von Zahn et al., 1988; Gerding et al., 2000; Chu et al., 2011). The capability of VAHCOLI to measure vertical and horizontal structures in metal densities allows to address several open science questions regarding the causes for the observed temporal and spatial variability (see section 5.5). The current version of the VAHCOLI-units is designed to
detect potassium atoms. In the future we envisage to develop new and compact lasers and/or frequency doubling techniques to measure other species, including iron and sodium.



### 2.5.4  Other

Several secondary parameters are typically derived from the prime observables such as the potential ($E_{pot}$) and kinetic energy ($E_{kin}$), momentum flux, and wave action densities. Note that the latter requires to measure the background mean winds in
order to consider the Doppler shifting effect on gravity waves. A Helmholtz decomposition of the flow, i. e., its divergent and rotational component, can be applied to better unterstand the physical processes involved. Lidars typically measure relative number densities, n(z), which allows to determine $E_{pot}$ from

$$E_{pot}^{mass} = \frac{1}{2} \cdot \frac{g^2}{N^2} \cdot \left( \frac{\Delta n}{n} \right)^2 \tag{2}$$

i. e., from number density instead of temperature fluctuations. This allows to reach to higher altitudes and avoids uncertainties
due to the start temperature.

Since VAHCOLI measures the dynamical and thermal components of the flow field, the heat flux due to fluctuations caused by gravity waves can also be derived. Statistical quantities are derived from fluctuations, for example longitudinal and transversal structure functions.

### 2.6  Lidar operation

After manufacturing, installation, and testing in the laboratory, the lidar can be transported to the location of interest where it is assembled for operation under field campaign conditions. The lidar is designed as a sealed and automated system, i. e., it is controlled remotely and can therefore run for long periods without any manual operation. This includes to stop measurements on short notice and very quickly (from one pulse to another) if required by, for example, air safety regulations or by bad weather conditions. Information regarding air safety is currently provided by an internal camera, and weather conditions are monitored
by an external weather station. Further constraints provided by external sources, e. g., a weather radar or air traffic control, can easily be incorporated into the lidar operation. If conditions are favorable again, the lidar switches on automatically within less than one minute.

## 3  First measurements

In the following we show results from the very first atmospheric measurements by a prototype of a VAHCOLI-unit ('first light')
performed in the period 17 to 19 January 2020. Some specifications of this lidar are summarized in Table 1. In Fig. 6 we show raw count rates observed on 19 January 2020 as detected by the detectors $D_{R-R}$ and $D_{Mie}$ (see Fig. 3). The goal of these measurements was to perform a first test of the entire lidar, i. e., laser, frequency control and analysis, telescope, detection system, Doppler free spectroscopy, lidar operation etc., under realistic conditions including rain, low temperatures, and storm, without touching the system for several days. Note that the FOV of the telescope was only $33\mu$rad which allowed measurements
even during full daylight. According to the description of the lidar presented in section 2.1.2 the confocal etalon is stabilized to a certain frequency, $\nu_{cf}$, and the power laser is normally tuned by typically $\pm1000$ MHz relative to this frequency. In the first


measurements presented here we concentrated on wind measurements (Doppler-Mie) and have therefore used a much smaller frequency range for tuning, namely only ±50 MHz. In the case shown in Figure 6, the etalon's central frequency, $\nu_{cf}$, was chosen such that it coincides with the mean resonance frequency of potassium to allow for a detection of the potassium layer.

The etalon transmits backscattered light from the atmosphere within a frequency range of $\nu_{cf} \pm \Delta\nu_{cf}$ where $\Delta\nu_{cf}$=FWHM/2 and FWHM∼7.5 MHz (blue line in Fig. 6). Note that the spectral width of the Mie peak is only ∼0.1 MHz which can be neglected in this context. Furthermore, $\Delta\nu_{cf}$ is much smaller than the spectral width of the Doppler broadened Rayleigh signal, i. e., only a very small fraction of the backscattered light from the atmosphere passes through the etalon, the rest is reflected and detected by a separate detector (red line in Fig. 6).

At altitudes below the potassium layer the total signal is due to backscattering from molecules (Rayleigh scattering) and a small contribution from Mie scattering from aerosols at altitudes below ∼30 km. When the frequency of the Mie peak is outside the frequency range of the confocal etalon, $\nu_{cf} \pm \Delta\nu_{cf}$, the backscattered light from the atmosphere detected at $D_{Mie}$ stems from Rayleigh scattering only, whereas $D_{R-R}$ detects Rayleigh (and resonance) scattering plus a small contribution from Mie scattering. The signal at $D_{R-R}$ is much larger compared to $D_{Mie}$ since most of the signal is reflected by the narrow band 295    confocal etalon (see Fig. 2 and 3). When the power laser frequency is within $\nu_{cf} \pm \Delta\nu_{cf}$, however, the signal at $D_{Mie}$ includes Mie scattering which varies when scanning the power laser, whereas the contribution from Rayleigh scattering is basically constant within $\nu_{cf} \pm \Delta\nu_{cf}$ because the Rayleigh peak is very flat within $\nu_{cf} \pm \Delta\nu_{cf}$. The signal at $D_{Mie}$ can therefore be used to measure Mie scattering only, which is subsequently used to subtract the Mie signal from the Rayleigh signal. Furthermore, the signal at $D_{Mie}$ is used to derive the Doppler shift of the Mie peak due to winds. In Figure 6 we show signals from the 300    detectors $D_{R-R}$ and $D_{Mie}$. The exponential decrease of the Rayleigh signal and some very small 'bumps' due to aerosol scattering are clearly visible. After subtracting the Mie contribution the signal can be used to determine a temperature profile (not shown). Note that temperatures can also be derived from the spectral width of the Rayleigh signal (not done in this first test). The Mie signal caused by stratospheric aerosols is roughly 0.1% – 10% of the Rayleigh signal and disappears above roughly 30 km.

In Fig. 7 we show line-of-sight winds derived from the Doppler shift of the Mie peak observed on 19 January with a height `Fig. 7` resolution of 200 m, integrated for a period of 20 minutes (18:10:00-18:30:00 LT). As can be seen from this Figure, the central frequency of the spectra changes with height, which is used to calculate line-of-sight winds. Note that the wind uncertainties shown in the right panel of Fig. 7 deviate substantially from our estimates presented later (see section 4.1) because the actual aerosol distribution is rather different and the lidar performance was not yet optimized. We compare these winds to ECMWF 310    (European Centre for Medium-Range Weather Forecasts) profiles which are closest in time and space (horizontal resolution: ∼9 km). The height resolution of ECMWF winds are approximately 250 m and 600 m at 5 km and 25 km, respectively. Here we have assumed somewhat arbitrarily that the lidar picks up 4% of the meridional winds corresponding to a off-zenith tilt of the laser beam by 2.3 degrees. This is certainly realistic when considering that no attempt has been made during this first test to exactly pointing the laser beam to the vertical. In the future the telescope pointing will be measured with high accuracy by 315    an integrated sensor.





The agreement between observations and ECMWF winds is very good considering the constraints regarding temporal/spatial coverage and sampling, and the fact that this was the very first test of the entire lidar using some preliminary optics. The results shown in Figure 7 demonstrate that the initial optical alignment of the lidar, including laser-beam adjustment relative to the telescope, was stable under harsh conditions and no re-alignment was required. The performance of the lidar during this first light measurements was significantly lower compared to expected future capabilities because the telescope and the detection system were not yet optimized. As will be explained in more detail in section 6 the efficiency of VAHCOLI-units will be improved further in the near term future.

## 4 Vertical/horizontal/temporal resolution and coverage versus accuracy

### 4.1 Expected performance

The following calculations of sensitivities and uncertainties are based on our experience with a potassium lidar which was housed in a container and operated in various remote locations such as on the research vessel Polarstern or in Spitsbergen (von Zahn and Höffner, 1996; Höffner and Lübken, 2007). In Fig. 8 we show expected count rates (R) as a function of altitude which in this case reaches the maximum possible value of R=$1\times10^7$ Hz at 20 km. We have assumed a laser power of 6 mJ (next generation of this laser) and an efficiency of the detector system of 30%. For a typical time and height interval of $\delta t$=5 min and $\delta z$=200 m, respectively, and a pulse repetition frequency of k=500/s this gives the number of photons as function of altitude according to equation 1, also shown in Fig. 8. For example, for R=$1\times10^7$ Hz (at 20 km) the number of photons (at 20 km) in a time and height interval of of 5 min and 200 m, respectively, is $N_{ph}$=$2\times10^6$. We have also indicated a typical dark count rate of 20 Hz in Figure 8 which is realistic for state-of-the-art detectors. The green line in Figure 8 gives the temperature uncertainties according to equation 5 (see later). The blue lines indicate the errors to measure winds from the shift of the Rayleigh spectrum (above 20 km) and from the shift of the Mie peak (below appr. 30 km). Hereby we have assumed that at 30 km the Mie signal is 0.5% of the Rayleigh signal increasing to 10% at 10 km. These values are consistent with typical observations of Mie scattering from stratospheric aerosols but may vary substantially throughout the season and from one location to another (Langenbach et al., 2019).

The calculation of the wind error is based on our experience that it takes approximately 100,000 photons to measure a wind with an accuracy of 1.35 m/s. Within limits (background noise etc.) the accuracy is proportional to the square-root of the number of photons. As can be seen in Figure 8, winds can be measured with high precision, i. e., better than 1 m/s below 40 km and 10 m/s below 70 km, respectively. Due to the small line width, Mie scattering is particularly suitable for measuring winds. In Figure 8 we also indicate the number of photons expected from an NLC layer assuming a backscatter coefficient at the peak of $\beta$=$30\cdot10^{-10}$/(m · sr) (see, for example, Fiedler et al., 2009). We also show typical backscattered signal from a potassium layer with a maximum number density of 50 atoms/cm$^3$.



## 4.2 Error analysis for Rayleigh temperatures

As is explained above, the lidars being built for VAHCOLI can measure the Rayleigh signal without contamination due to aerosols. We consider altitudes sufficiently below the uppermost height where uncertainties due to the start temperature are negligible. Starting from an altitude bin centered at $z_1$ with a temperature $T_1$ and number density $n_1$, the following equation

gives the temperature error in the next height bin (at $z_2$) due to uncertainties in density measurements $\Delta n_1$ and $\Delta n_2$ at level $z_1$ and $z_2$, respectively:

$$\Delta T_2 = \exp\left(-\frac{z_2 - z_1}{H_p}\right) \cdot \frac{n_1}{n_2} \cdot T_1 \cdot \sqrt{\left(\frac{\Delta n_1}{n_1}\right)^2 + \left(\frac{\Delta n_2}{n_2}\right)^2} \tag{3}$$

where $n_2$ is the number density in the altitude bin centered at $z_2$, and $H_p$ is the pressure scale height. This equation can be further simplified by assuming that $H_p \approx H_n$ within a height interval of $\delta z = z_2 - z_1$ (which is typically a few hundred meters

only) and that the uncertainties in $n_i$ are determined by Poisson statistics of counting $N_i$ photons, i. e.

$$\frac{\Delta n_i}{n_i} = \frac{\Delta N_i}{N_i} = \frac{1}{\sqrt{N_i}} \quad ; \quad (i=1,2) \tag{4}$$

Since $N_1 \approx N_2$ within the height interval $\delta z$ we finally get:

$$\Delta T \approx T \cdot \sqrt{\frac{2}{N(z)}} \tag{5}$$

The number of photons counted per time and altitude interval, $N(z)$, decreases with altitude according to

360 $$N(z) = N_{ref} \cdot \left(\frac{z_{ref}}{z}\right)^2 \cdot e^{-(z-z_{ref})/H_n} \tag{6}$$

where $N_{ref}$ is the number of photons counted at the reference altitude $z_{ref}$, and $H_n$ is the number density scale height. In Fig. 9 altitude profiles of temperature errors according to equation 5 are shown assuming a number of photons at the reference level (20 km) of $N_{ref}=2\times10^6$ (see above) and $N_{ref}=2\times10^5$, respectively. Since count rates are normally suppressed at lower altitudes (to avoid a saturation of detectors) they may be increased at higher altitudes, for example by reducing the attenuation

in the receiver. Using telescopes with appropriate diameters is another method to focus on certain altitude ranges. In Fig. 9 we have assumed an enhancement of N due to 'cascading' by a factor of 100 (for $N_{ref}=2\times10^6$) at altitudes above 50 km which leads to a reduction of $\Delta T$ by a factor of 10. In total, typical temperature errors are smaller than 5 K up to the upper mesosphere. Another method to increase the effective count rate is to increase the height range ($\delta z$) and/or the integration time ($\delta t$). In Fig. 10 the effect of increasing $\delta t$ and/or $\delta z$ on temperature errors is shown. More precisely, temperature errors ($\Delta T$) are shown as

a function of g = $(\delta t \cdot \delta z)/(\delta t_{ref} \cdot \delta z_{ref})$ where $\delta t_{ref}$=5 min and $\delta z_{ref}$=200 m, and the number of photons at the reference level

Fig. 9

Fig. 10





($z_{ref}$=20 km) is $N_{ref}$=2·10$^6$. For example, increasing $\delta z \cdot \delta t$ by a factor of 60 (e. g., by increasing the integration time from 5 min to 1 hour and the height interval from 200 m to 1 km) decreases the temperature error at 50 km from $\Delta$T=5.3 K (g=1) to $\Delta$T=0.7 K (g=60).

### 4.3   Multi-beam operation and horizontal coverage

The flexibility of VAHCOLI allows to place the lidars at distances which are optimized according to the science objectives (see below). The current plan is to build four VAHCOLI-units (N$_V$=4) with five beams (N$_B$=5) each, where one beam is pointing vertically and 4 beams are pointing at a zenith angle of e. g. $\chi$=35° in two orthogonal directions. In Fig. 11 (upper panel) an example of a constellation of four VAHCOLI-units with five beams each is shown. At any given altitude there are a total of $N_L$=5×4=20 laser beams available, which gives a total of $\sum_{i=1}^{N_L-1} i$=190 combinations of horizontal distances. In the lower

panel of Fig. 11 these 190 horizontal distances are shown (abscissa) for a selection of 12 different scenarios (ordinate), including the scenario shown in the upper panel of Fig. 11. Horizonal scales from several kilometers up to hundreds of kilometers are detectable where nearly all directions in the horizontal plane are covered. Even quasi-equal horizontal distances contain rather important information since they are located at different places (non-homogeneity) and/or point in different directions (non-isotropy).

If the beams in all four lidars are aligned equidistantly the maximum horizonal coverage is (N$_V$×(N$_B$-2) -1)×$\Delta$x(z) = 11×$\Delta$x(z), where $\Delta$x(z)= z·tan($\chi$) is the (altitude dependent) horizonal distance between two beams. For example, for $\chi$=35° the horizontal distance between two beams at 50 km is 35.0 km and the maximum horizontal distance covered by 4 lidars which are located at equal horizontal distances is 385 km. Many other scenarios may be chosen, for example concentrating on smaller scales by placing the systems very close to each other and choosing a small zenith angle with similar azimuths. On

the other hand, several VAHCOLI-units may be located at distances of several hundred kilometers to concentrate on processes on synoptic scales. Any combination of such scenarios may be chosen, of course, depending on the availability of lidars and appropriate locations.

## 5   Science capabilities

### 5.1   General

Dynamical processes on medium spatial scales (up to several hundred km) are important for the atmospheric energy and momentum budgets which are directly relevant for climate models on regional and global scales (see, e. g. Becker, 2003; Shepherd, 2014). More specifically this concerns the question how energy and momentum are transferred from large to small scales (or vice versa?) and how the horizontal/vertical transport of energy, momentum, and constituents are described correctly. A prominent example of dynamical impact on the background atmosphere is the summer mesopause region at high latitudes

where temperatures deviate by up to 100 K from a state which is controlled by radiation only. This strong deviation is primarily caused by gravity waves which deposit energy and momentum and lead to a 'residual circulation' and related upwelling and

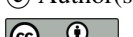



cooling. Major aspects of this dynamical control of the atmosphere are only poorly understood due to the complexity of the problem, both from the experimental and theoretical point of view. In models, the impact of these processes is typically considered by parameterizations. If, or if not, these parameterizations adequately describe the real atmosphere can best be

verified by comparing models with observations which are capable of fully characterizing the atmospheric variability at these medium scales. Temporal and spatial variability is observed in the atmosphere at a large range of scales which reflect various processes and their (mostly) non-linear interactions. Since these fluctuations vary in time and space it is necessary to measure spatial and temporal variations of, e. g., winds and temperatures simultaneously to achieve a complete picture.

      The ultimate aim of VAHCOLI is to characterize the three-dimensional and time-dependent morphology of atmospheric

flow, including gravity waves. This allows to disentangle temporal from spatial variability of the main flow and associated fluxes and to test frequently assumed simplifications in modeling (and some observations) regarding homogeneity, isotropy, and stationarity. In this paper we concentrate on medium scales, i. e., horizontal distances of one to few hundred kilometers, and vertical distances of 100 m to several kilometers.

      It is often assumed that atmospheric processes on medium scales are stationary which is very unlikely to be true in general

since energy and momentum are continuously removed from the flow. If, or if not, this assumption is perhaps valid within certain limits or within certain scales may be verified by comparing with suitable observations spanning a sufficient range of temporal and spatial scales. Other assumptions include isotropy and homogeneity, for example regarding fluctuations in zonal and meridional direction. Again, such similarities are rather unlikely because normally the background flow is systematically different in zonal compared to meridional directions.

There are several rather fundamental questions in atmospheric dynamics where VAHCOLI can contribute to a better understanding. For example, are the governing processes of fluctuations at spatial scales larger than the buoyancy scale ($L_b$, see below) determined by saturation of breaking gravity waves or by un-isotropic large scale turbulence being damped vertically by buoyancy, or by a combination of both? Which type of instability is most relevant in a specific situation, velocity shears (Kelvin-Helmholtz instabilities) or convective instabilities ? Another fundamental aspect of atmospheric variability regards the

question if spectra are separable, i. e., if they can be expressed in the form

$$A(\omega, k_x, k_y, k_z) \overset{?}{=} A_0(\omega) \cdot A_1(k_x) \cdot A_2(k_y) \cdot A_3(k_z) \tag{7}$$

      Separability is frequently assumed to be valid but there is no fundamental reason why this should be the case (Fritts and Alexander, 2003). VAHCOLI aims at contributing to a better understanding of these fundamental aspects with observations of winds and temperatures with substantial temporal and spatial resolution and coverage.

In atmospheric science the variability of winds and temperatures is frequently characterized by a spectral index $\xi$ in the expression k$^\xi$ (k = wavenumber). For example, zonal winds ($u$) as a function of horizontal wavenumber ($k_h$) in the range from few to several hundred kilometers follow a quasi-universal law $u(k_h) \sim k_h^{-5/3}$ (Nastrom and Gage, 1985). Keeping the temporal and spatial variability of the atmosphere in mind it may require several days of averaging to actually observe such a behavior (Weinstock, 1996). Furthermore, measuring $\xi$ alone may not be sufficient to characterize the underlying physical





process unambiguously. For example, gravity waves and stratified turbulence (see below) may exhibit the same spectral behavior in a specific situation, although the fundamental concepts are very different. In any case, the spectral representation should include as many observables as possible (zonal/meridional/vertical winds, temperatures, kinetic/potential energies, wave action density, momentum flux, etc.) in terms of vertical/horizontal wavenumbers and frequencies.

A powerful tool to describe atmospheric flows is to apply a Helmholtz decomposition, namely to separate the kinetic energy
of the flow into divergent and rotational components:

$$E_{kin}(k) = E_{rot}(k) + E_{div}(k) \tag{8}$$

This sometimes allows to distinguish different physical processes from each other (see below). Obviously, this requires 3d-observations of the flow. Furthermore, since this separation may vary in time, a time-resolved measurement of the entire horizontal wind vector is required, as is planned for VAHCOLI.

## 5.2 Gravity waves

Lidars have frequently been applied to measure gravity waves, both in case studies and also deducing climatologies (see Hauchecorne and Chanin, 1980; Liu and Gardner, 2005; Rauthe et al., 2008; Kaifler et al., 2015; Chu et al., 2018; Baumgarten et al., 2017; Strelnikova et al., 2021, for some examples). More recently, lidars have been applied to simultaneously detect GW in temperatures and winds in the middle atmosphere and to apply hodograph methods which allows to derive potential and kinetic
energy and to separate upward and downward propagation (Baumgarten et al., 2015; Strelnikova et al., 2020). Note that background winds are needed to determine Doppler shifting which is essential, for example, to unambiguously separate upward and downward progression of gravity waves, where the latter could for example be due to secondary wave generation (Kaifler et al., 2017; Becker and Vadas, 2018). Sometimes only certain parts of the GW field are measured and dispersion and polarization relations are applied (plus further assumptions regarding isotropy and/or stationarity) to derive quantitative results (Ern et al.,
2004; Pautet et al., 2015).

To exploit the capabilities of VAHCOLI of studying gravity waves, we concentrate on waves with medium frequencies, i. e., $\hat{\omega} \gg f \sim 10^{-4}$/s at mid latitudes ($\hat{\omega}$ = intrinsic frequency, f = Coriolis parameter). Corresponding periods are smaller than roughly 17 h and, of course, larger than the Brunt-Väisälä (BV) period of several minutes. Since $E_{div}/E_{rot} = \hat{\omega}^2/f^2 \gg 1$ this implies that the divergent part of the GW flow is much larger compared to the rotational component.
The dispersion relation for gravity waves (assuming that $\lambda_z \ll 4\pi \cdot H$) is

$$\frac{\hat{\omega}}{N} = \sqrt{\frac{k_x^2 + \frac{f^2}{N^2} \cdot k_z^2}{k_x^2 + k_z^2}} \approx \frac{k_x}{\sqrt{k_x^2 + k_z^2}} = \cos(\varphi) \text{ for } N \gg f \tag{9}$$

where N is the Brunt-Väisälä frequency, and $\varphi$ is the angle between the phase propagation direction and the horizontal direction (see Dörnbrack et al., 2017, for a recent summary on lidar applications for atmospheric GW detection). For intrinsic periods significantly larger than the Brunt-Väisälä period (but still smaller than f), we have $\hat{\omega}/N \ll 1$, i.e., $\varphi \sim 90°$, i. e., the



phase progression is nearly vertical. We note, that lidars are also capable of detecting GW with larger periods, i. e. inertia gravity waves, both in winds and temperatures (Baumgarten et al., 2015).

A graphic representation of the dispersion relation is shown in Fig. 12. Several investigations have studied the specifics of GW which normally propagate from low to high altitudes including the question which part of these GW can be observed by satellites (see, for example, Preusse et al., 2008; Alexander et al., 2010). We do not include radiosondes and balloons here

due to their sporadic nature and limited height coverage. Satellites can only observe GW with typical horizontal/vertical wavelengths larger than appr. 50-100 km and 3–5 km, and periods larger than typically 1-2 hours. However, the effect of high frequency waves on the circulation is crucial since the vertical flux of horizontal pseudo-momentum is given by $F_P = \overline{u'w'} \cdot \overline{\rho} \cdot (1 - f^2/\hat{\omega}^2)$ which is largest for mid and high frequency gravity waves, e. g., when $\hat{\omega} \gg f$ (Fritts and Alexander, 2003). As can be seen from Figure 12 VAHCOLI covers an important part of the gravity wave spectrum which is not accessible by

satellites, in particular waves with small horizontal wavelengths and small periods (large frequencies). As mentioned before, the phase of these waves preferentially propagates vertically, e. g., the energy propagates obliquely.

The aim of VAHOCLI is to characterize the three-dimensional time-dependent morphology of gravity waves. A comprehensive characterization of gravity wave propagation requires to measure the three-dimensional vector of phase propagation, i. e., the vertical and the horizontal components. Horizontal fluxes of gravity wave momentum are typically ignored (compared

to vertical) in middle atmosphere modeling where it is often assumed that the effect of gravity waves takes place directly above the source and instantaneously. It is known from model studies, however, that GW can propagate over large horizontal distances before depositing momentum and energy (Alexander, 1996; Ehard et al., 2017; Stephan et al., 2020). Furthermore, a background varying with time can change the propagation of GW (e. g., by refraction) and can drastically modify the deposition of momentum and its effect on the background flow (Senf and Achatz, 2011). Simulations of GW propagation show

that the horizontal distance between wave packets usually increases with altitude (see, for example, Alexander and Barnet, 2007). This is favorable for VAHCOLI since the horizontal distance between obliquely pointing beams also increases with altitude (see Figure 11). Furthermore, it is known that the spatial and temporal distribution of gravity wave sources influences their effect on middle atmosphere dynamics (Šácha et al., 2016). A more fundamental question addresses the role of non-linear interactions of gravity waves compared to a quasi-linear superposition. This leads to rather different concepts regarding grav-

ity wave parametrization (Lindzen, 1981; Gavrilov, 1990; Fritts and Lu, 1993; Medvedev and Klaassen, 1995; Hines, 1997; Becker and Schmitz, 2002). It could well be that the applicability of one concept or the other depends on the temporal/spatial scales under consideration. In order to measure and study the effects outlined above it is obviously necessary to observe gravity waves in all directions over a longer period of several hours or even days, and with sufficient horizontal coverage. Such instrumental capabilities are envisaged for VAHCOLI.

## 5.3   Stratified turbulence (ST)

The concept of stratified turbulence (ST) has recently been developed to explain the energy cascading in stratified flows at mesoscales as an alternative to classical linear or non-linear breakdown of gravity waves. This transfer is relevant for the momentum and energy budgets which affect the Lorenz cycle and thereby (regional) climate modeling. Lindborg (2006) has



developed an energy cascade theory for these scales in a strongly stratified fluid which involves horizontal and vertical length
scales as well as kinetic and potential energy. The theory of ST has recently been applied to wind measurements by radars in
the mesopause region (Chau et al., 2020).

ST resembles the well-known energy spectra (horizontal kinetic energy and potential energy) characterized by $k_h^{-5/3}$
(Nastrom and Gage, 1985). This theory invokes strong non-linearities (in contrast to 2D-turbulence and to weakly nonlin-
ear interacting gravity waves) and the cascading of energy from large to small scales (see, for example Billant and Chomaz,
2001; Lindborg, 2006; Brethouwer et al., 2007; Lindborg, 2007, and references therein). It covers horizontal scales smaller
than synoptic scales ($L_h$) and larger than buoyancy ($L_b$) and Ozmidov ($L_O$) scales, and it covers vertical scales between $L_b$
and $L_O$. The Ozmidov scale $L_O=\sqrt{\epsilon/N^3}$ describes the largest scales in classical isotropic Kolmogorov turbulence which are
not effected by buoyancy (N=~0.02/s ; $\epsilon$ = energy dissipation rate of turbulence). The buoyancy scale $L_b=u_h/N$ ($u_h$= typical
horizontal velocities of ST) characterizes the largest vertical scale of stratified turbulence, whereas $L_h= u_h^3/\epsilon$ is the largest
horizontal scale of ST structures. Variability at scales larger than $L_h$ are related to large scale processes dominated by the
Coriolis force. Introducing typical Kolmogorov isotropic turbulence velocities, $u_t=\sqrt{\epsilon/N}$, several relationships can be de-
rived, such as $L_h/L_b=(u_h/u_t)^2$ and $L_h/L_O=(u_h/u_t)^3$. The magnitude of the outer scale of ST ($L_h$) is typically a few hundred
kilometers (see, for example Avsarkisov et al., 2021).

Order of magnitude estimates for velocities and time constants related to these scales are derived from expressions such as
$u_h= \sqrt[3]{L_h \cdot \epsilon}$ and $\tau_h= L_h/u_h$. Applying typical values, namely N=0.02/s (BV period = 5min), $\epsilon$=100 mW/kg, and $L_h$=100 km
results in the following order-of-magnitude values: $L_b$=1.01 km, $L_O$=0.112 km, $u_h$=21 m/s, $u_t$=2.24 m/s, $u_h/u_t$=9.6, and
$\tau_h$=77 min, respectively. A collection of relationships and representative values is presented in Table 2. A more detailed
representation of spatial and temporal scales as well as velocities associated with stratified turbulence is shown in Fig. 13 for
a large range of $\epsilon$-values. Note that most quantities depend on season, latitude, and altitude (regarding $\epsilon$ see, for example,
Lübken, 1997). Some dimensionless numbers are frequently used to characterize the relevance of physical processes. For
example, the horizontal Froude number, which is the ratio of inertial to buoyancy forces, must be small to allow for ST to exist:
$Fr_h \ll 1$. Note that $Fr_h= u_h/(N \cdot L_h) = L_b/L_h= \epsilon/(N \cdot u_h^2)$. Indeed $Fr_h$ is very small for the examples shown in Table 2.
Another relevant parameter is the buoyancy Reynolds number, $Re_b= \epsilon/(\nu \cdot N^2)$, which should be large both for ST and for
the Kelvin-Helmholtz instability regime ($\nu$ = kinematic viscosity). In the height range from the lower stratosphere to the upper
mesosphere, and turbulence intensities of $\epsilon$=10 mW/kg and $\epsilon$=100 mW/kg this parameter varies between $Re_b$=2.5×10$^5$ to
$Re_b$=25 and $Re_b$=2.5×10$^6$ to $Re_b$=250, respectively, i. e., $Re_b$ is indeed much larger than unity. Regarding the application of
VAHCOLI we note that the requirements to cover ST scales, namely a vertical/horizontal resolution of 200m/2km, a horizontal
coverage of up to 200 km, and temporal and velocity resolutions of 10-20 min and 0.5-1 m/s, respectively, are well within the
instrumental capabilities of VAHCOLI. The temporal development of the flow is important to judge various forcings, energy
injection, the conversion of $E_{pot}$ and $E_{kin}$, and the transition to stationary conditions (see, e.g., Lindborg, 2006).

There are several aspects of ST theory which are particularly relevant for a comparison with observations by VAHCOLI.
For addressing the question, if energy is cascading from large to small scales ('forward') or the other way around ('inverse')
it is helpful to consider not only the horizontal kinetic energy spectra (typically from aircraft observations) but also the ver-

Tab. 2

Fig. 13





tical spectra of horizontal kinetic and potential energy, typically from balloon borne observations (Li and Lindborg, 2018;

Alisse and Sidi, 2000; Hertzog et al., 2002). Note that a fundamental scale invariance of the Boussinesq equations in the limit of strong stratification implies an equi-partitioning of potential and kinetic energy (Billant and Chomaz, 2001). Regarding spectra, the ST theory (invoking downscale energy flow) predicts that vorticity $\Phi(k)$ and divergence $\Psi(k)$ spectra should be of similar magnitude, $\Phi(k) \approx \Psi(k)$, whereas for spectra dominated by gravity waves one would expect $\Phi(k) \ll \Psi(k)$, and for stratified turbulence dominated by vortical coherent structures one expects $\Phi(k) \gg \Psi(k)$. Furthermore, it is helpful to measure

spectra of longitudinal and transversal velocity structure functions simultaneously (Lindborg, 2007).

In summary, the expected horizontal and vertical coverage of the flow field by VAHCOLI will allow to study details of the relationship between rotational and divergent components of mesoscale dynamics including the important question, how energy is transfered from large to small scales. The instrumental capabilities of VAHCOLI will cover spatial and temporal scales being highly relevant for mesoscales. Apart from the Helmholtz decomposition there are other important quantities,

such as the helicity, H=$\boldsymbol{v} \cdot rot(\boldsymbol{v})$, which may be helpful to separate vortical coherent structures from GW and to characterize the flow and its potential impact on the background atmosphere (Marino et al., 2013). Again, such a comprehensive analysis requires a 3d-characterization of the flow field, as is envisaged for VAHCOLI.

## 5.4 Other dynamical parameters

There are several dynamical processes in the atmosphere which take place at spatial or temporal scales which are normally out-

550 side the range of VAHCOLI, at least for the time being. For example, the smallest scales of inertial range turbulence are on the order of $L_\eta = (\nu^3/\epsilon)^{1/4}$. Measuring fluctuations at $L_\eta$ scales offers a unique chance to unambiguously determine $\epsilon$ (Lübken, 1992). However, $L_\eta$ varies by several orders of magnitude from the troposphere to the upper mesosphere and is in the range of centimeters to several meters only. It will be challenging to detect fluctuations at these scales by, for example, placing several VAHCOLI-units very close to each other. On the other hand, measuring the longitudinal and transversal structure functions of

555 winds and temperatures at somewhat larger scales also allows to derive reasonable estimates of $\epsilon$. Furthermore, we envisage to measure the spectral broadening of the stratospheric aerosol signal to an extent that allows to deduce turbulent velocities. Note that typical turbulent velocities are on the order of 1 m/s (see Table 2) which corresponds to a spectral broadening of the Mie peak of 2.5 MHz. This is much larger than the Doppler broadening of the Mie peak due to Brownian motions (roughly 0.1 MHz).

Trace constituents may sometimes be used as passive tracers for transport and mixing. This mainly concerns vertical and horizontal advection and mixing of stratospheric aerosols and noctilucent clouds, but also the transport of metal atoms. Care needs to be taken when interpreting such measurements since these constituents may not be passive tracers, i. e., they may experience modifications, for example, by variable background temperatures. In the future we envisage to measure small scale turbulence (see above) and to improve the spatial resolution of aerosol observations to an extend that the eddy correlation

technique to measure turbulent transport should be applicable.

On the other side of the spectrum of scales, tides are global scale phenomena with horizontal wavelengths of several hundred kilometers. Certainly, the relevant periods and vertical wavelengths are within the scope of standard MLT lidars



(see Baumgarten et al., 2018, for a recent example). Regarding horizontal wavelengths, one could consider placing several VAHCOLI-units at very large distances.

Dynamical phenomena are frequently characterized by calculating statistical quantities, such as the variance and higher moments (skewness, kurtosis, etc.) as well as intermittency. Due to the operational advantages of VAHCOLI (low cost, unattended operation, low infrastructure demands, long-term stability etc.) there is an opportunity to extend such an analysis to bi- or multi-variate distributions, for example, correlating wind components at various places with each other, or with temperatures.

### 5.5 NLC, PSC, background aerosols, and metal densities

Ice layers in the summer mesosphere at middle and polar latitudes are known as NLC ('noctilucent clouds') (Gadsden and Schröder, 1989). They exhibit a large range of temporal/spatial variability which can even be observed by naked eye or by camera. Most of these variations are presumably related to gravity waves, tides, and associated instability processes (see Baumgarten and Fritts, 2014, for a more recent example). NLC are studied in great detail by modern lidars which sometimes detect temporal fluctuations on time scales down to seconds, or other unexpected characteristics (Hansen et al., 1989; Alpers et al., 2001; Gardner et al., 580 2001; Kaifler et al., 2013). NLC are frequently used in models describing dynamical phenomena such as gravity wave breaking (Fritts et al., 2017). This raises the question, up to which scales NLC can be treated as passive tracers. Note that several processes act on similar temporal and spatial scales, for example, nucleation, sedimentation, and horizontal transport. Furthermore, there is an impressive amount of observations of mesospheric ice clouds available from satellites, which sometimes show unexpected temporal and/or spatial variations ('voids') (see Russell III et al., 2009, for details on a more recent satellite mission 585 dedicated to NLC science). Understanding the physics of NLC is important, for example, to interpret long term variations of ice layers and their potential relationship to climate change (Thomas, 1996; von Zahn, 2003; Lübken et al., 2018). Similar science questions occur regarding PSC, which also play a crucial role in ozone chemistry. Very thin layers of background aerosols have been observed in the stratosphere which are presumably caused by intrusion of mid-latitude air into the winter polar vortex (see, for example, Plumb et al., 1994; Langenbach et al., 2019). Several VAHCOLI-units could be placed at appropriate 590 locations, e. g., at the edge of the polar vortex, to observe the temporal and spatial development of such intrusions.

For solving some of the open science questions regarding NLC, PSC, and background aerosols, it is very helpful to distinguish between temporal and spatial (horizontal) variations, and to know the status of the background atmosphere. VAHCOLI is designed to detect these aerosol fluctuations and to observe background temperatures and winds simultaneously by applying high resolution spectral filtering (see section 2.5.2).

Despite substantial progress in recent years, the physics and chemistry of metal layers still leaves many open questions, for example, regarding their (meteoric) origin, their spatial and seasonal distribution, the impact of diffusion and turbulent transport, as well as the effect of gravity waves and tides on number density profiles (see Plane, 2003, for a recent review on mesospheric metals). The morphology of metal profiles offers a variety of phenomena on short spatial and temporal scales, such as sudden (sporadic) layers and their connection to ionospheric processes, or the uptake of metal densities on ice particles 600 (see, i.e., Hansen and von Zahn, 1990; Alpers et al., 1993; Collins et al., 1996; Plane et al., 2004; Lübken and Höffner, 2004).





Many of these anomalies can best be studied by distinguishing temporal from spatial variations. VAHCOLI is designed to observe metal layers (potassium for now) at various locations with high time resolution.

## 6    Outlook and Conclusion

Several improvements regarding the technical performance of lidars being used for VAHCOLI are currently in progress or are foreseen for the near term future. This concerns, for example, the optical layout of the telescope, the development of multi-beam operation, and the output of the power laser. A power increase of up to several Watt applying ring-laser technology is currently under development, even without employing a separate amplification stage. The VAHCOLI-units are designed to be extended to further wavelengths, for example by installing a second seeder laser and a SHG (second harmonic generation) to simultaneously apply resonance scattering on potassium ($\lambda$=770 nm) and iron ($\lambda$=386 nm), all during night and day. Stimulated Raman emission may also be considered for producing laser light in the infrared.

The robust and compact design of the VAHCOLI-units and their stand-alone operation capability allows to consider somewhat extraordinary operations, for example on air planes, balloons, ships or trains, or at remote places such as Spitsbergen or Antarctica. Some of these applications have been proven to be realistic, however so far with a substantial effort regarding technical realization, man-power, and costs (von Zahn et al., 1996; Höffner and Lübken, 2007; Lübken et al., 2017; Chu et al., 2018). VAHCOLI may also be extended to other wavelengths being relevant for thermospheric or space applications (Höffner et al., 2018; Munk et al., 2018; Höffner et al., 2019). Due to its compact design and autonomous operation a VAHCOLI-unit may also be of interest for measurements from satellites (Strotkamp et al., 2019). The final aim is to further improve the VAHCOLI-units and to develop a cost-effective multi-purpose lidar where several systems may be employed at various locations, being operated quasi-autonomously.

In summary, we have presented the VAHCOLI concept which consists of a cluster of lidars ('units') to study the middle atmosphere in four dimensions, namely high temporal, vertical, and horizontal resolution and coverage. The concept relies on the development of a new type of lidar which is compact ($\sim$1m$^3$), can be operated stand-alone during night and day (even under harsh field conditions), and still offers a performance which is comparable to, or even better than existing lidars. The innovative approach for this lidar is based on very fast and high spectral resolution spectroscopy. Apart from a narrow band spectral filter in the detector system (confocal etalon: 7.5 MHz) the key component of the new lidar is a newly developed diode-pumped alexandrite laser which offers a small line-width of $\sim$3.3 MHz which is significantly better compared to most lasers currently being used in lidars (e. g. Nd:YAG and lasers pumped by Nd:YAG: $\sim$50–100 MHz). The laser is flexible and can be tuned quickly over, for example, a Doppler broadened line originating from atmospheric Rayleigh or resonance scattering. At the same time the laser can also cover a large frequency range and may be used for other absorption lines. The flexibility of the lidar allows to concentrate and optimize the performance regarding certain height ranges or atmospheric parameters, for example, measuring winds from stratospheric aerosols or temperatures from resonance scattering in the upper mesosphere/lower thermosphere. For the first time, temperatures and winds from Rayleigh and/or resonance scattering can be deduced in the presence of aerosols (stratospheric aerosols, noctilucent clouds). The compact layout of the lidar reduces the





requests on optical alignment. Since these lidars allow autonomous operation under harsh field conditions at remote locations,
they are ideally suited for the VAHCOLI concept, namely to employ a cluster of these units to simultaneously cover atmospheric
parameters in the vertical and horizontal direction with sufficient spatial and temporal resolution. The units shall be cost-
effective using off-the-shelf components and 3d-printing of mechanical subsystems. In this paper we have discussed some
relevant science applications of such observations, for example regarding gravity waves and stratified turbulence. We have
presented measurements of a first prototype of such a lidar which demonstrates the suitability of the new lidar for VAHCOLI.
This paper has addressed just a selection of the numerous future application scenarios of VAHCOLI.

*Data availability.* Data are available at: https://www.radar-service.eu/radar/en/dataset/cZfWBGffZIhILdHs?token=xjEfCgYJrPfxWwnksuxQ

*Author contributions.* Josef Höffner designed the lidar and provided some of the data shown in the paper, Franz-Josef Lübken prepared the
manuscript (with contributions from JH) and some of the plots.

*Competing interests.* The authors declare that they have no conflict of interest.

*Acknowledgements.* The contribution, support, and expertise of the Fraunhofer Institute for Laser Technology (ILT) in Aachen when develop-
ing a new laser is highly appreciated. Alsu Mauer, Jan Froh, and Thorben Mense have contributed substantially to the technical development
of the lidar being used in VAHCOLI. We thank Victor Avsarkisov, Gerd Baungarten, Erich Becker, Koki Chau, and Michael Gerding for help-
ful discussions and GB for providing the ECMWF data. This work was supported by the Leibniz SAW project FORMOSA (K227/2019), by
the DFG project PACOG (LU-1174/8-1) which is part of the MSGWaves research group, and by the BMWi project ALISE (FKZ 50RP1605).





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

870



| | |
|---|---|
| laser power | 1 W |
| repetition rate | 500 Hz |
| laser energy per pulse | 2 mJ |
| laser pulse length | 780 ns (234 m) |
| laser beam profile | $M^2$=1.1 |
| laser beam divergence | ~10-15 $\mu$rad |
| telescope field-of-view | 33 $\mu$rad |
| $\Delta\nu$ of power laser | 3.3 MHz |
| $\Delta\nu$ of seeder laser | ~0.1 MHz |
| wavelength | 769.898 nm ; $K(D_1)$ |
| $\Delta\nu$ of interference filter | 150 GHz |
| $\Delta\nu$ of broad band etalon | 1000 MHz |
| $\Delta\nu$ of confocal etalon | 7.5 MHz |

$\Delta\nu$ = spectral width (FWHM)

**Table 1.** Specifications of the prototype lidar being used for the first measurements in January 2020.

**Tables**





|  |  |  | $L_h$=100 km | | $L_h$=400 km | |
|---|---|---|---|---|---|---|
|  |  |  | $\epsilon$=10 | $\epsilon$=100 | $\epsilon$=10 | $\epsilon$=100[1] |
| $L_\eta$ | = | $(\nu^3/\epsilon)^{1/4}$ | 0.100 | 0.056 | 0.100 | 0.056 |
| $L_O$ | = | $\sqrt{\epsilon/N^3}$ | 35 | 112 | 35 | 112 |
| $L_b$ | = | $u_h/N$ | 500 | 1077 | 793 | 1709 |
| $L_h$ | = | $u_h{}^3/\epsilon$ | | is given above | | |
| $u_t$ | = | $\sqrt{\epsilon/N} = L_O \cdot N$ | 0.71 | 2.24 | 0.71 | 2.24 |
| $u_h$ | = | $\sqrt[3]{L_h \cdot \epsilon} = L_b \cdot N$ | 10.0 | 21.5 | 15.9 | 34.2 |
| $\tau_O$ | = | $L_O/u_t = 1/N$ | 0.83 | 0.83 | 0.83 | 0.83 |
|  | = | $\tau_b = L_b/u_h = 1/N$ | | | | |
| $\tau_h$ | = | $L_h/u_h = L_h/(L_b \cdot N)$ | 167 | 77 | 420 | 195 |
|  | = | $(u_h/u_t)^2 / N$ | | | | |
| $L_O/L_\eta$ | = | $Re_b{}^{3/4}$ | 354 | 1988 | 354 | 1988 |
| $L_b/L_O$ | = | $u_h/u_t = 1/Fr_h{}^{1/2}$ | 14 | 10 | 22 | 15 |
| $L_h/L_b$ | = | $(u_h/u_t)^2$ | 200 | 93 | 504 | 234 |
|  | = | $u_h{}^2 \cdot N / \epsilon = 1 / Fr_h$ | | | | |
| $L_h/L_O$ | = | $(u_h/u_t)^3 = 1/Fr_h{}^{3/2}$ | 2826 | 894 | 11304 | 3574 |
| $Re_b$ | = | $\epsilon/(\nu \cdot N^2)$ | 2500 | 25000 | 2500 | 25000 |

[1] $\epsilon$ in mW/kg. All lengths in m, all velocities in m/s, all time constants in minutes.

**Table 2.** Length scales, typical velocities, time constants, and various ratios of length scales relevant for (stratified) turbulence. The scales are shown for two cases of $L_h$, namely $L_h$=100 km and $L_h$=400 km, as well as for two cases of energy dissipation rates $\epsilon$, namely $\epsilon$=10 mW/kg and $\epsilon$=100 mW/kg. For the kinematic viscosity a value of $\nu$=0.01 m²/s was chosen which is typical for an altitude of 45 km (note that $\nu$ increases exponentially with increasing height). See text for more details.



**Figures**

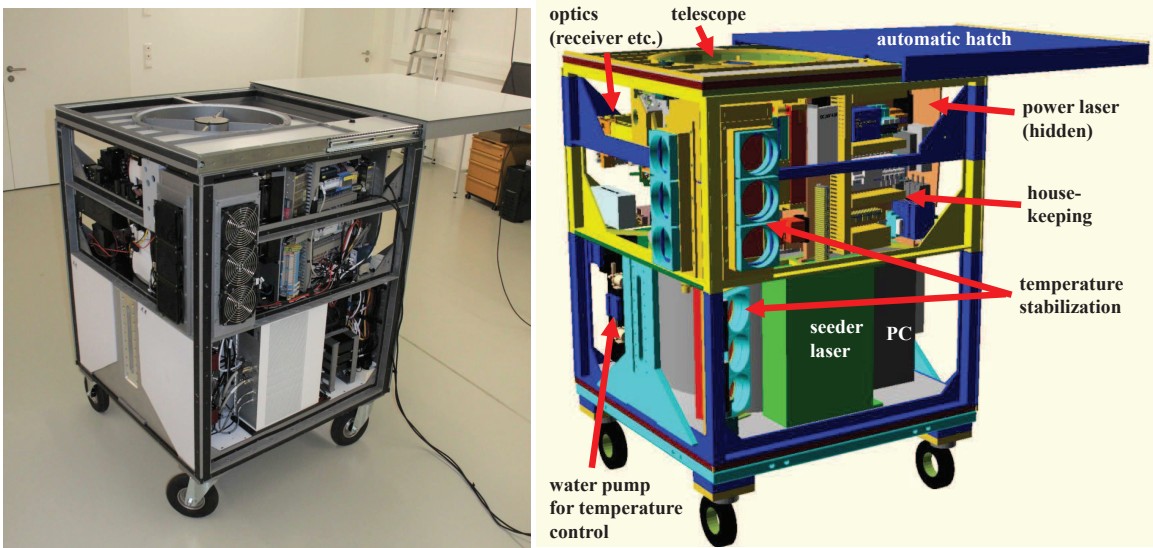

**Figure 1.** Foto and technical drawing of a lidar being used for VAHCOLI. Some important parts can be recognized, such as the telescope, the seeder laser, and the PC. Temperature control and stabilization (different for different sections) is realized using air ventilation and a water cooling system. The dimensions of a VAHCOLI-unit (without the wheels and with closed hatch) are 96 cm x 96 cm x 110 cm (length x width x height). The weight is approximately 400 kg, and the power consumption 500 Watt under full operation.



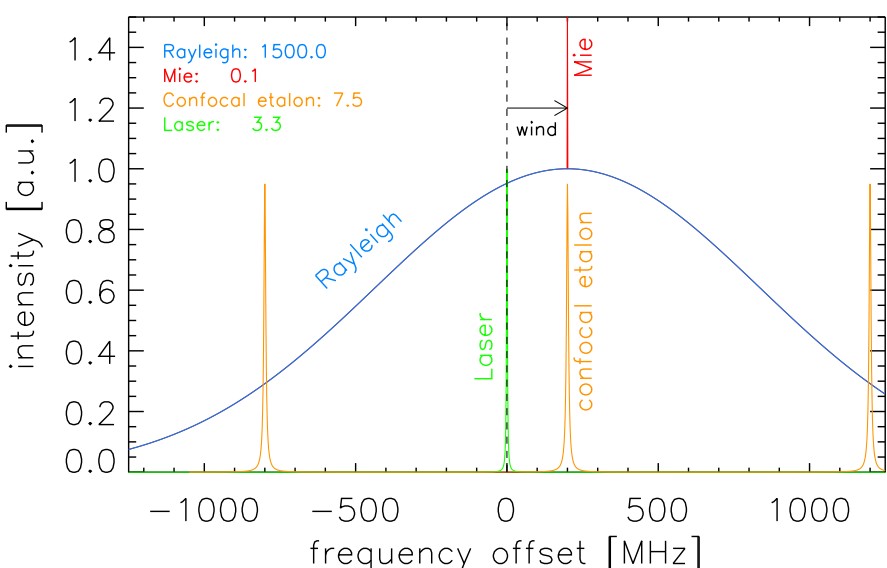

**Figure 2.** Schematic of typical spectral widths (FWHM) relevant for VAHCOLI. The spectral widths (in MHz) are given in the insert. Blue: Doppler broadened Rayleigh signal (same order of magnitude for resonance scattering), red: Mie scattering by aerosols, green: spectrum of the power laser, orange: spectrum of the confocal etalon (free spectral range: 1000 MHz), in this case centered at the Mie peak. Doppler shifting by background winds leads to a shift of approximately 2.6 MHz for a wind of 1 m/s.





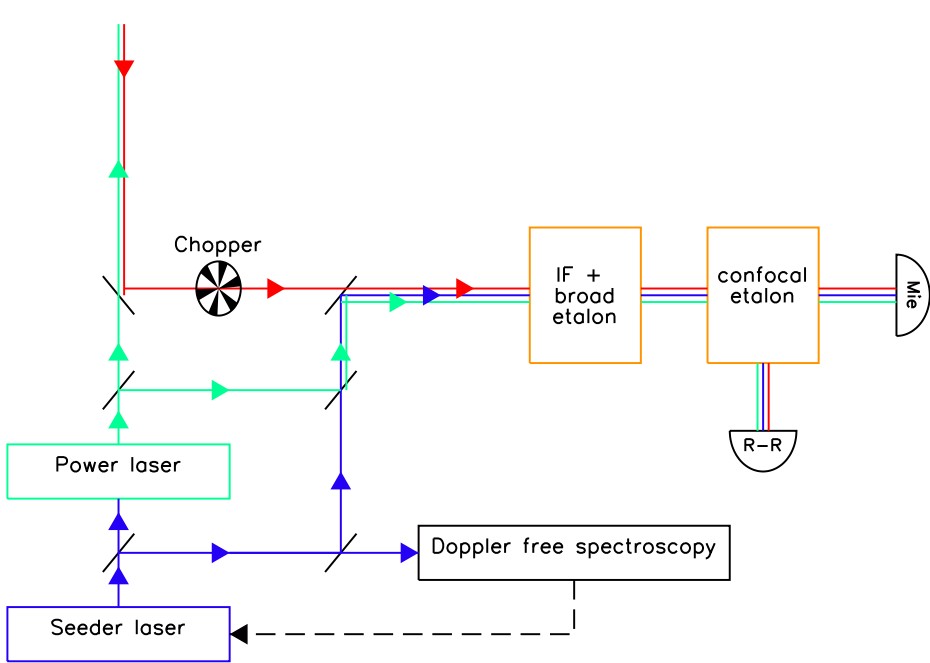

**Figure 3.** Sketch of a lidar being used for VAHCOLI. The frequency of the power laser (green) is controlled by a seeder laser (blue) which itself is controlled by high precision Doppler free spectroscopy. When the chopper is open, the signal from the atmosphere (red) is fed into the receiver system which consists of a broad band interference filter (IF, $\Delta\nu$=150 GHz), a broad band etalon ($\Delta\nu$=1000 MHz), a narrow band confocal etalon ($\Delta\nu$=7.5 MHz), and detectors for the Rayleigh/resonance ('R-R') and Mie channels. When the chopper is closed, parts of the seeder and power lasers are fed into the receiver system to measure their frequencies.



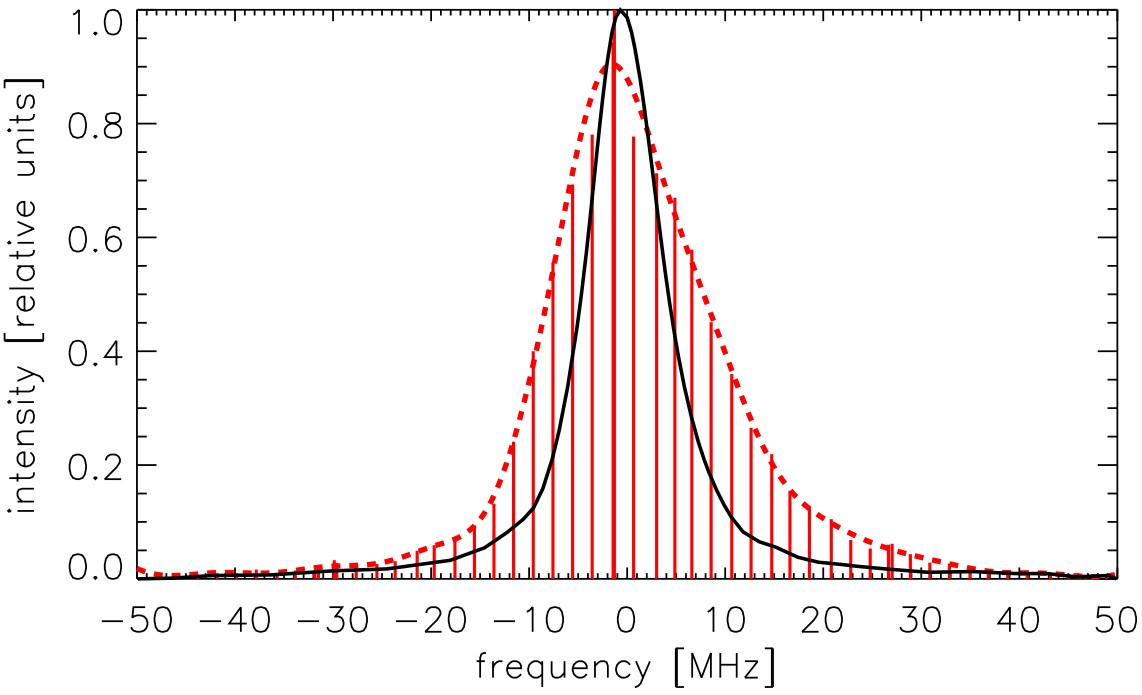

**Figure 4.** Measurement of the convolution of the spectra of the power laser (FWHM~3.3 MHz) and the confocal etalon (FWHM ∼ 7.5 MHz) taken for a period of ∼5 minutes (150,000 pulses) during first light operation in January 2020. The power laser is tuned over the spectrum of the confocal etalon where a total of 50 individual mean frequencies of the power laser with a difference of 2 MHz each are chosen (vertical red lines). The power laser matches the requested mean frequencies to better than ∼100 kHz, which is within the thickness of the red lines. The dashed red line is an approximate envelope of the vertical red lines. The black line is the spectrum of the confocal etalon measured separately by tuning the seeder laser over the spectrum of the confocal etalon. The slightly asymmetric shape of the red dashed line is a result of a non-perfect optical alignment.



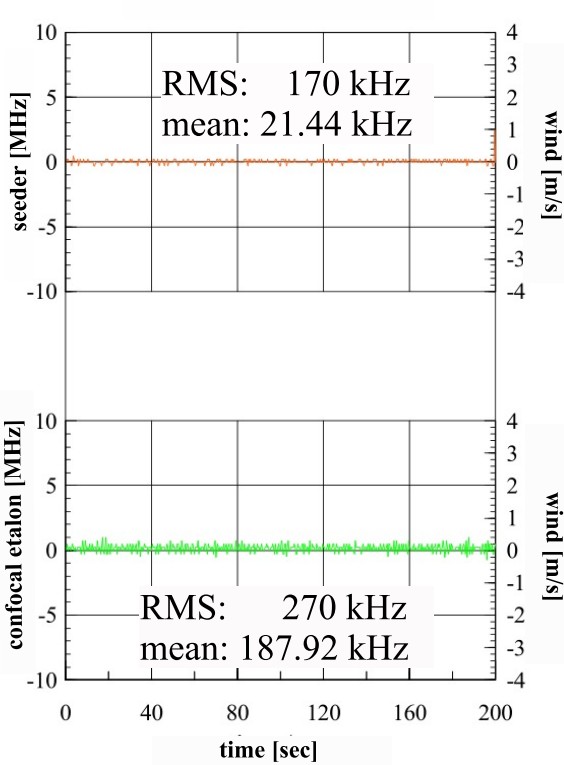

**Figure 5.** Upper panel: Frequency of the seeder laser determined with a time resolution of 1/10 second. More precisely the differences between the nominal frequency and the actual frequencies are shown, where the latter are determined by comparing with high precision Doppler-free spectroscopy. Note that the seeder was tuned up and down within a range of 100 MHz and several scans are averaged within a time period of 1/10 s. The mean of the frequency difference is 21.44 kHz, and the RMS of the fluctuations is 170 kHz. Lower panel: same, but for the confocal etalon. The seeder laser from the upper panel is fed into the confocal etalon and the nominal frequency is compared to the actual (seeder laser) frequency. The mean of the frequency difference is 187.92 kHz, and the RMS of the fluctuations is 270 kHz. For comparison, note that a wind speed of 0.1 m/s corresponds to a frequency shift of 260 kHz.

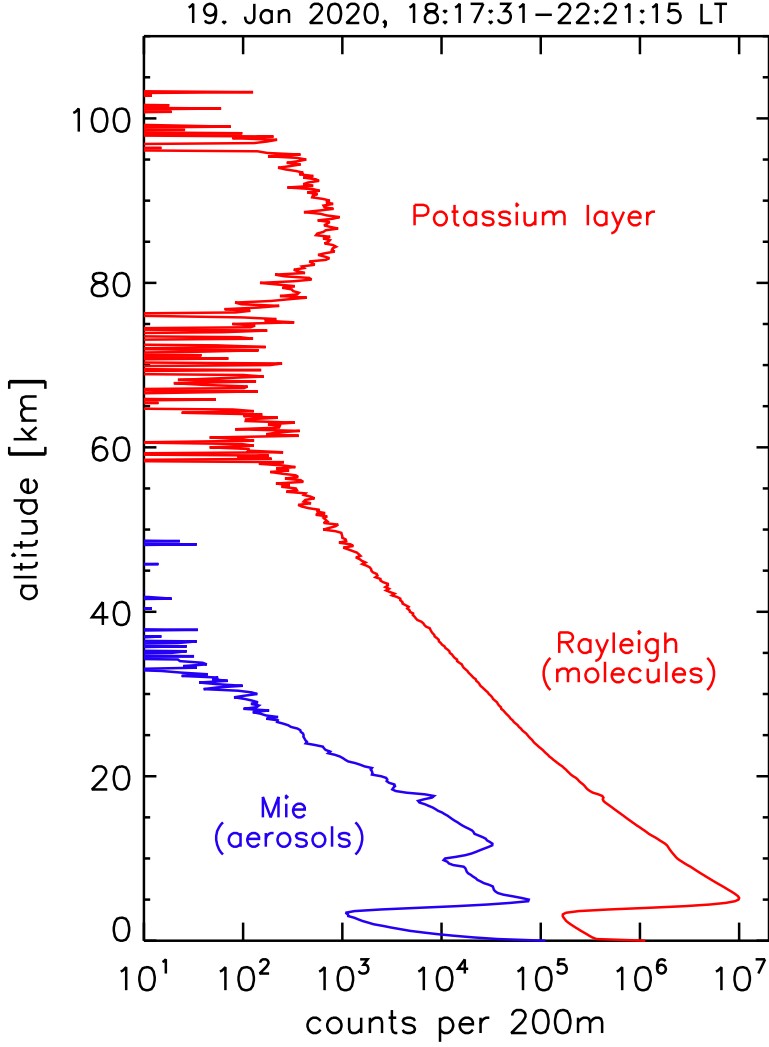

**Figure 6.** Altitude profiles of raw backscattered signals ('first light') observed by the detectors $D_{R-R}$ and $D_{Mie}$ observed on 19. January 2020, 18:17:31–22:21:15 LT (see Fig. 3). Red line ($D_{R-R}$): mainly Rayleigh scattering on molecules (below ∼70 km) and resonance scattering on potassium atoms (∼75–100 km). Blue ($D_{Mie}$): Mie scattering on stratospheric aerosols. Below ∼30 km the signal at $D_{R-R}$ includes a very small contribution from Mie scattering which is subtracted during further processing (see text for more details). The decrease of the signals below approximately 5 km is caused by the chopper, blocking the atmospheric signal.



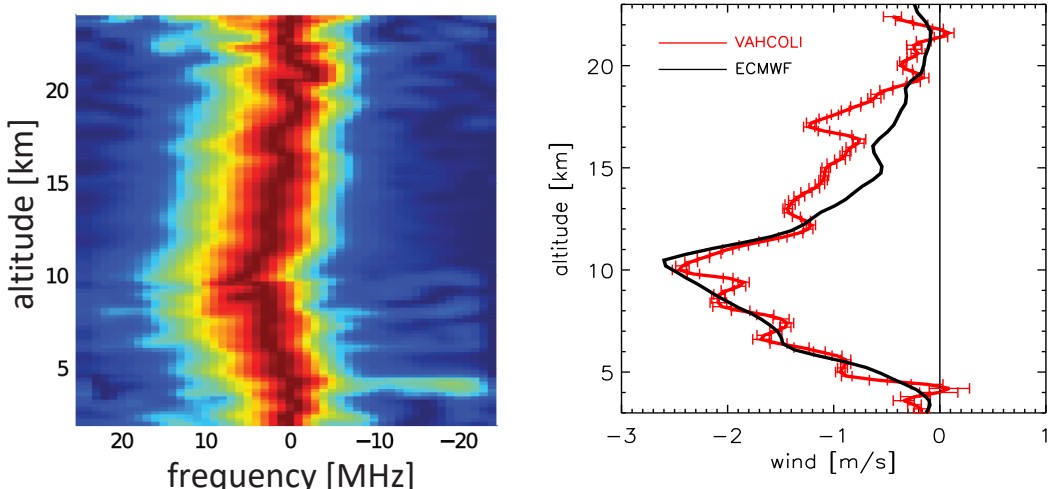

**Figure 7.** Wind profiles from a prototype of a VAHCOLI-unit as derived from the Doppler shift of the Mie signal. Left panel: spectra of the Mie signal (relative to the mean frequency $\nu_o$) as a function of altitude. The shift of the spectra relative to $\nu_o$ is used to derive line-of-sight winds. Right panel: observed line-of-sight wind profile (red line and error bars) with a height resolution of 200 m, integrated in the time period 18:10:00–18:30:00 LT on January 19, 2020. Black line: ECMWF wind profile closest in time and space. A small fraction (4%) of ECMWF meridional winds has been added to ECMWF vertical winds assuming that the lidar has a small tilt relative to the vertical by 2.3 degrees. See text for more details.



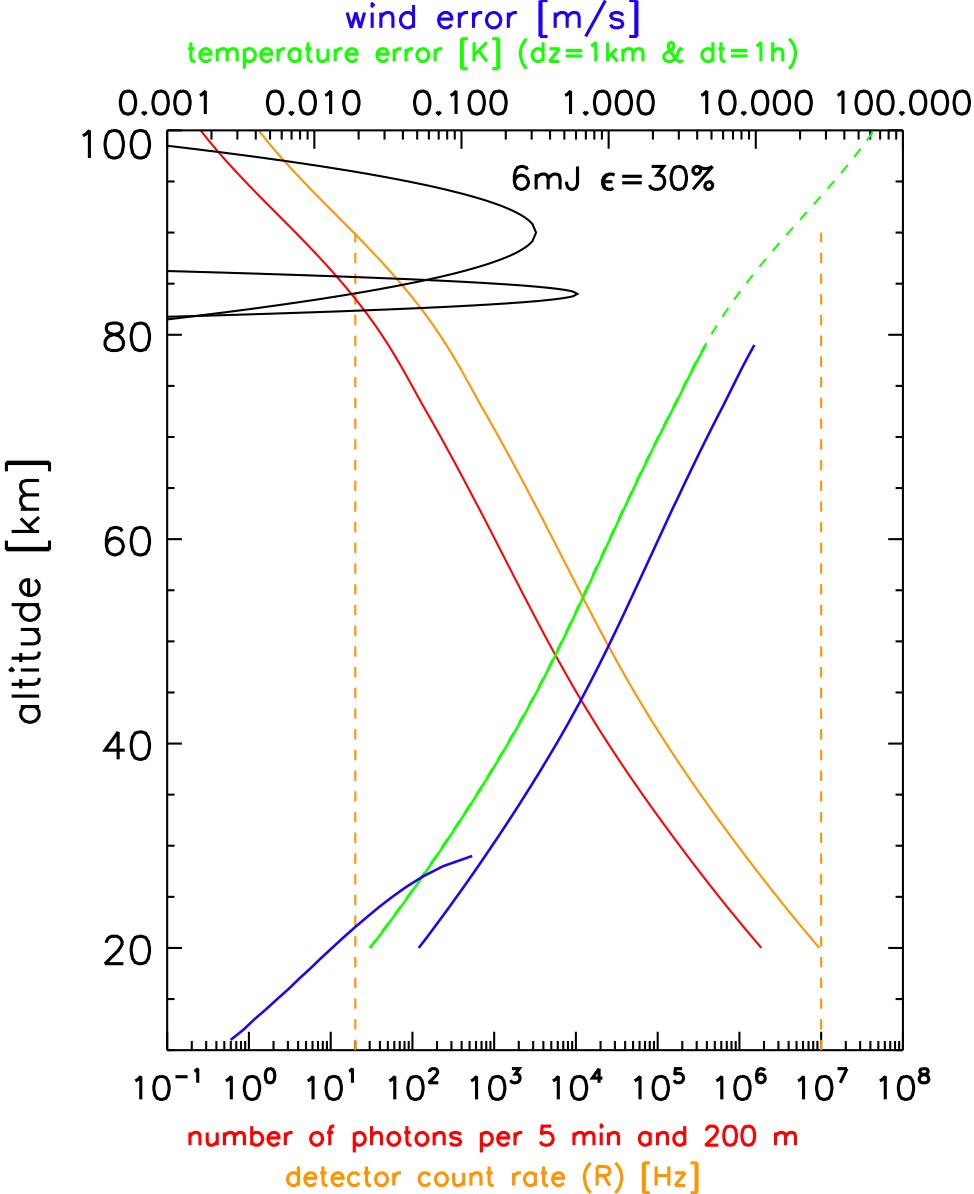

**Figure 8.** Sensitivity to measure temperatures, winds, and aerosols. Lower abscissa, orange: count rate; red: number of photons ($N_{ph}$) per time and height interval of $\delta t$=5 min and $\delta z$=200 min, respectively. $N_{ph}$ would increase by a factor of 60 if instead $\delta t$=1 hour and $\delta z$=1 km had been chosen. Black lines at 80-100 km: approximate number of photons expected from the K and NLC layers. The vertical orange lines indicate the maximum achievable count rate ($10^7$ Hz) and typical dark count rates of the detector (20 Hz). Upper abscissa: green: temperature error for an integration time and altitude bin of $\delta t$=1 hour and $\delta z$=1 km, respectively. Blue: error of winds obtained from stratospheric aerosols (Doppler-Mie, lower part) and from Doppler-Rayleigh (upper part). These calculations assume a laser energy of 6 mJ and an overall detection efficiency of 30%.





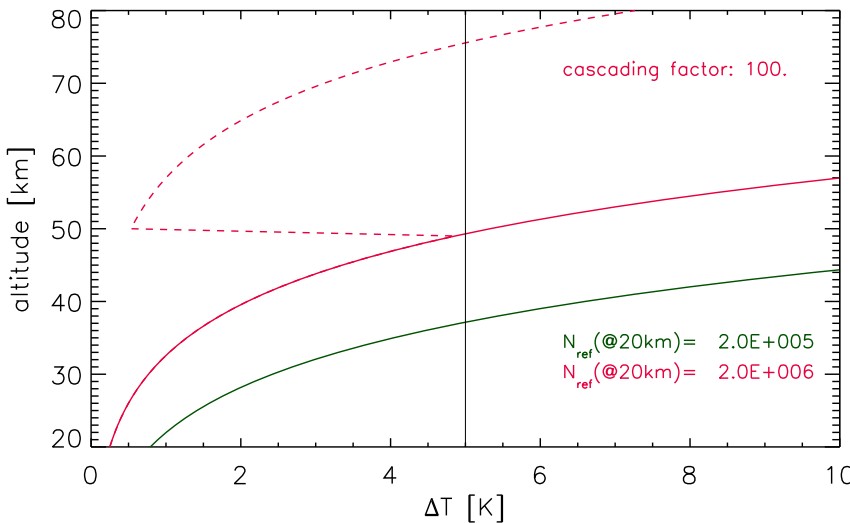

**Figure 9.** Temperature error according to equation 5. Here we have assumed that the number of photons counted at 20 km altitude per time-interval $\delta t$=5 min and height-interval $\delta z$=200 m is $2 \cdot 10^6$ (red) and $2 \cdot 10^5$ (green), respectively. The dashed red line indicates the effect of cascading the detector by a factor of 100 (see text for more details).

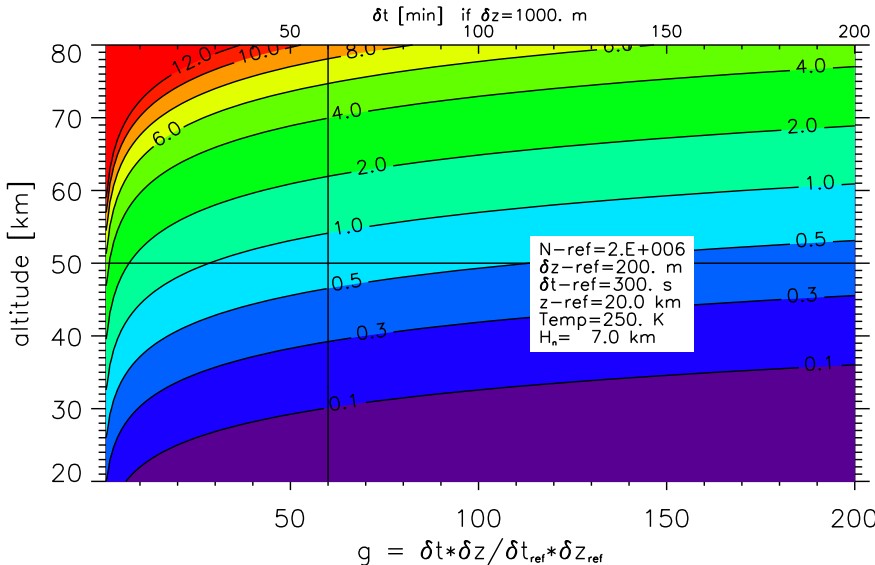

**Figure 10.** Impact of increasing the time ($\delta t$) and/or the height interval ($\delta z$) on the temperature error from equation 5. Temperature errors are shown as a function of g = $(\delta t \cdot \delta z)/(\delta t_{ref} \cdot \delta z_{ref})$ where $\delta t_{ref}$=5 min and $\delta z_{ref}$=200 m. $N_{ref}$=$2 \cdot 10^6$ is the number of photons at $z_{ref}$=20 km. The upper abscissa shows the time interval $\delta t$ in minutes if $\delta z$=1000 m.





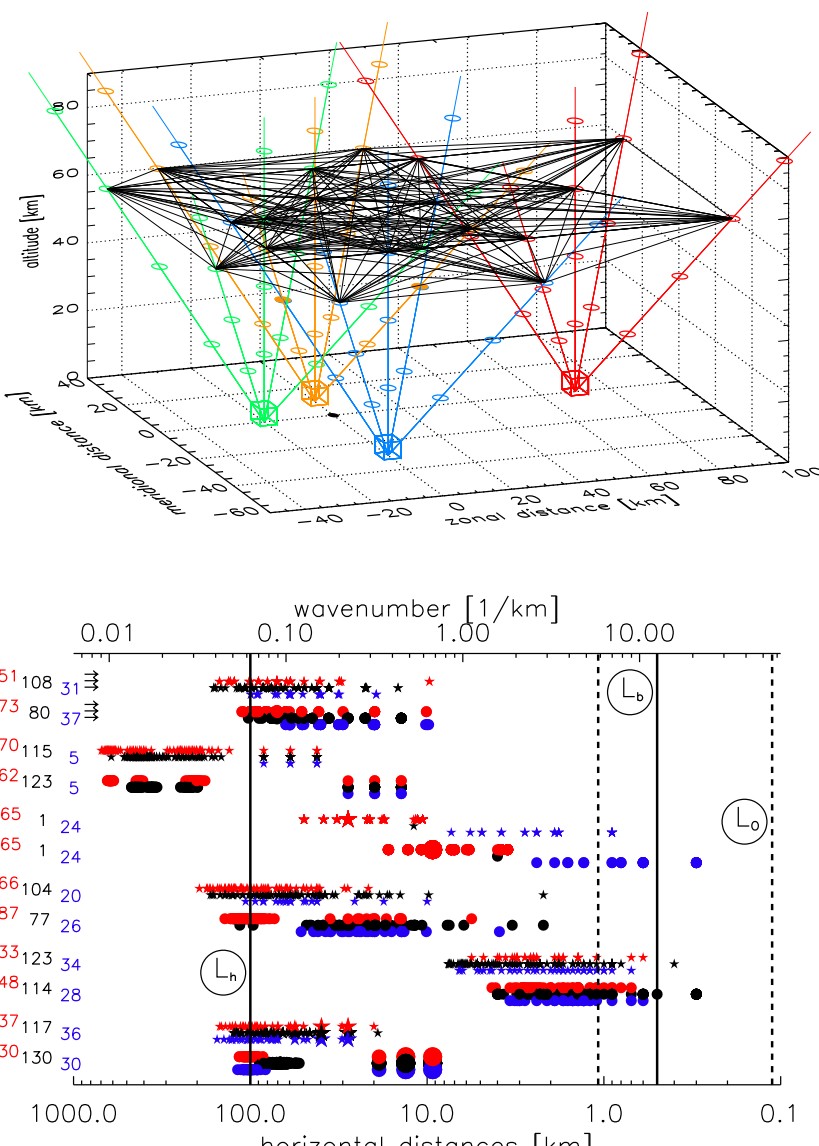

**Figure 11.** Horizontal coverage by VAHCOLI consisting of 4 units with 5 beams each. Upper panel: example of positions of 4 lidars (with 5 beams each) located at horizontal positions (x,y) of (0/-30, blue), (-20/0, green), (0/10, orange), and (70/0, red), all in km. At each location the zenith angles of the 5 beams are $0°$, $35°$, $35°$, $35°$, and $35°$. The azimuth angles of the 4 tilted beams at all 4 stations are $45°$, $135°$, $225°$, and $315°$. The locations of the beams at certain altitudes, namely 20 km, 40 km, 60 km, and 80 km, are marked by small circles. Thin black lines show all horizontal connections between the circles at 60 km. Lower panel: Horizontal coverage by 4 lidars with 5 beams each, where the 4 locations are chosen from various scenarios. Each point gives the horizontal distance between two beams at an altitude of 20 km (circles) or 60 km (stars). The size of the symbols indicates that several identical distances are represented. The colors indicate the azimuth of these connections: red: east-west ($0°±20°$), blue: north-south ($90°±20°$), black: others. The corresponding number of cases are listed on the left ordinate (total of 190). The uppermost scenario (highlighted by small arrows) shows the case presented in the upper panel. The vertical lines indicate scales which are relevant for stratified turbulence (see section 5.3 for more details).

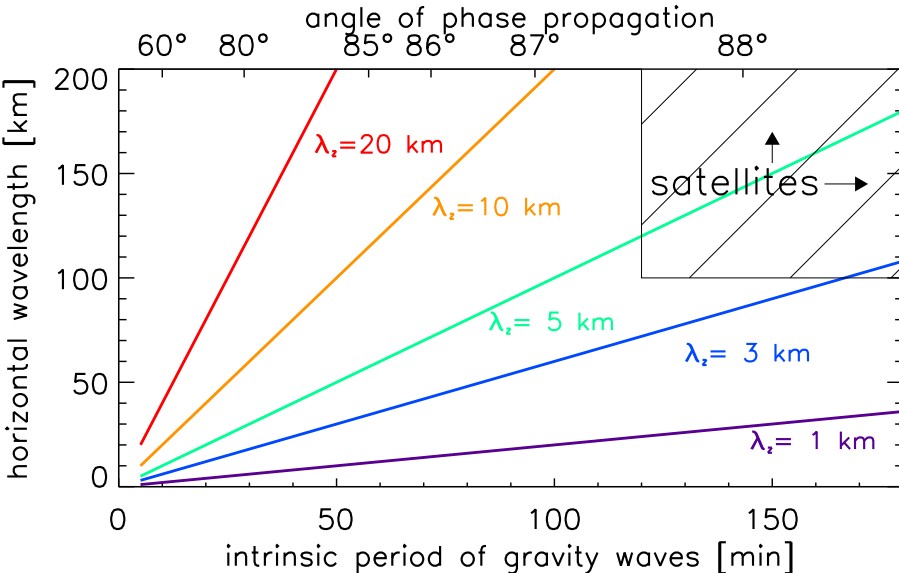

**Figure 12.** Scales of gravity waves relevant for VAHCOLI deduced from the dispersion relation of gravity waves for mid frequencies (equation 9). Horizontal wavelengths are shown as a function of intrinsic period for various vertical wavelengths (colored lines). The angle of phase propagation relative to the horizontal direction is given on the top abscissa. The hatched area (top right) indicates the lower part of ranges covered by satellites.



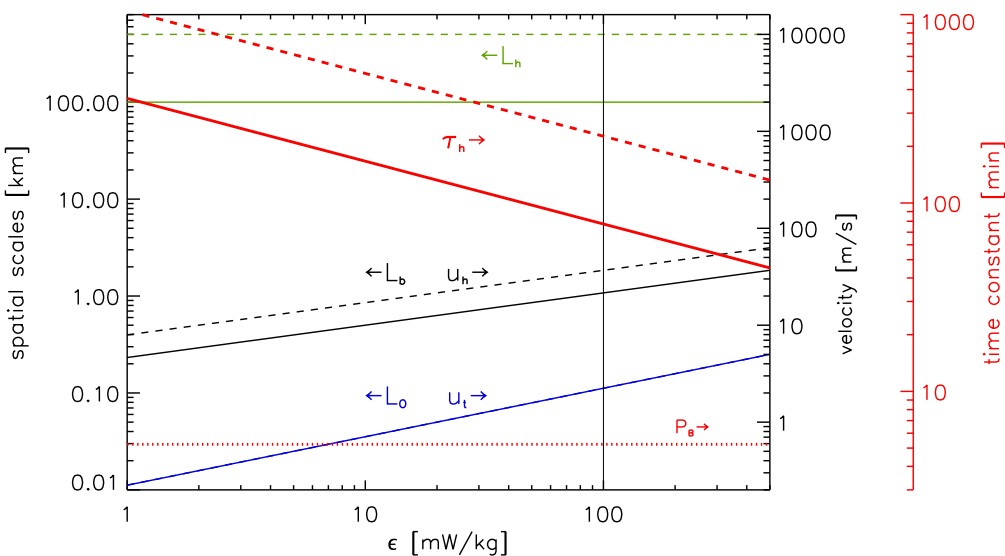

**Figure 13.** Typical scales relevant for stratified turbulence (ST) as a function of turbulent energy dissipation rate ($\epsilon$). Length scales (left axis): largest horizontal scales of ST ($L_h$, green), buoyancy scales ($L_b$, black), and Ozmidov scales ($L_O$, blue). Typical time scales ($\tau_h$, right red axis) and horizontal velocities ($u_h$, right black axis) related to $L_h$ are shown, as well as typical turbulent velocities ($u_t$). The red dotted line indicates a typical Brunt-Väisälä period ($P_B$). All scales are shown for two cases, namely $L_h$=100 km (solid lines) and $L_h$=400 km (dashed lines). The scales are defined in section 5.3. See also Table 2.