# Peer review of "VAHCOLI, a new concept for lidars: technical setup, science applications, and first measurements"

_Atmospheric Measurement Techniques, 2021_

## Author Comment (AC1)

**Reply to reviewer's comments on the paper**

**'VAHCOLI, a new concept for lidars: technical setup, science applications, and first measurements'**

submitted to AMT by Franz-Josef Lübken and Josef Höffner

Manuscript Number: amt-2021-33

**Reviewer #1:**

We appreciate the encouraging and positive comments from the reviewer. We have taken the suggestions for improvements into account when preparing the revised version of the manuscript. We have marked the changes and respond to the reviewer's comments point by point in the following. In this response we repeat the reviewer's comments in blue, put our comments in green, and mark the modified version of the text in the manuscript in red. We have polished the text somewhat with the help of a native English speaker (shown in cyan in the manuscript).

*„An excellent, well-written and readable paper on an autonomous, small, versatile, precise and robust lidar instrument. The instrument is an excellent idea, and it uses many different state-of-the-art and even cutting-edge lidar techniques. The manuscript describes the method and a prototype, presents a sample dataset and elaborates well on the kind of questions in atmosphere research that can be studied with an arrray of a reasonable number of such instruments. The text, figures, tables and equations are useful, clear and easy to understand.*"

1. „Somewhere, for instance near the end of the Introduction, it would be fair to make reference to the following publication by Kaifler and Kaifler (2020), which describes a similar small, precise, robust and autonomous lidar instrument: https://doi.org/10.5194/amt-2020-418 . A sentence or two about similarities and differences between VAHCOLI and CORAL would be useful."

   We agree with the reviewer and have added a reference to this paper and also a short comment regarding the differences to VAHCOLI. Apart from not being able to measuring winds and not being daylight capabable, CORAL does not allow for an optical separation of scattering from molecules and aerosols.

   „More recently, a compact and autonomous Rayleigh/Mie/Raman lidar has been developed for middle atmosphere research which, however, cannot measure winds nor be operated during daylight (Kaifler and Kaifler, 2021).

2. „Line 169-173: It does not become clear why no active control of the outgoing laser beam is necessary. Perhaps one important piece of information was not described, or the formulation is not clear enough."

   Indeed, this is an important advantage of VAHCOLI and is finally related to the

[0] last updated: 16th April 2021     c:/Papers-FJL/AMT/2021-VAHCOLI/Reply-AMT-2021.tex     (F.-J. Lübken)

1-38 compact design of the entire lidar which preserves the alignment between the out-
1-39 going laser beam and the optical axis of the telescope. Once adjusted, no correction
1-40 on short time scales is needed since the photons scattered by 180° follow the same
1-41 optical path as the outgoing photons. Slow drifts are compensated for by a control
1-42 loop maximizing the return power. We have expanded the explanation in the text.

1-43 „... This is achieved as follows: Once the outgoing laser beam and the optical axis
1-44 of the telescope are co-aligned the photons being scattered by 180 degrees from the
1-45 atmosphere follow the optical path of the outgoing laser-beam but in the retrograd
1-46 direction and thereby arrive at the detectors. The light from the atmosphere is
1-47 separated from the outgoing laser pulse using its polarization characteristics. The
1-48 compact design of the lidar ensures that the alignment between the laser beam and
1-49 the telescope is preserved on short timescales, i. e., no active control of the outgoing
1-50 laser beam on a pulse-to-pulse basis is needed. Slow drifts of the laser beam relative
1-51 to the optical axis of the telescope caused by, for example, temperature drifts are
1-52 compensated for by a control loop (maximizing the atmospheric signal) with a time
1-53 constant of few minutes."

1-54 3. „Consider adding, for instance in section 2.4, information about the chopper rotation
1-55 frequency and beam diameter at the chopper, which determines the opening time
1-56 near 5 km in Figure 6."

1-57 It turns out that the role of the chopper in i) blocking atmospheric light from short
1-58 distances and stray light within VAHCOLI as well as ii) synchronizing the firing of
1-59 the power laser is rather complex. This concerns, for example, the position of the
1-60 Gaussian laser beam relative to the chopper blades and the accuracy of the laser
1-61 firing and the stability of the chopper rotation. We decided to synchronize the chop-
1-62 per to the laser, and not vice versa as is done in most lidars. A description of the
1-63 technically details of the chopper is beyond the scope of this paper. We have added
1-64 a short note on the chopper in section 2.2 of the revised version.

1-65 „The chopper shown in Fig. 3 helps to separate backscattered light from the middle
1-66 atmosphere from other sources, e. g., stray light within VAHCOLI. The rotation
1-67 speed of the chopper and the open segments within the chopper are chosen to ef-
1-68 fectively open the detectors for atmospheric light at an altitude above 3 km and to
1-69 allow for the firing of 500 laser pulses per second. The opening of the chopper is
1-70 synchronized to the firing of the power laser."

1-71 4. „Line 220 and perhaps several other places in the text: Consider adding 'line-of-sight
1-72 winds'.

1-73 We have added 'line-of-sight winds' in line 220, and also in other places in the text.

1-74 5. „Line 225: consider adding 'to derive metal atom number densities' or similar.'
1-75 We have added 'metal atom number densities' in line 225.

1-76 6. „In section 3, Table 1, and Figs. 6-7, consider adding where these measurements were
1-77 performed. From lines 163-165, the reader can guess that this was Kühlungsborn,
1-78 but the information might be useful near the figures."

1-79 Yes, the measurements were performed at the location of IAP in Kühlungsborn. We
1-80 have added the location at the beginning of section 3 and also in Table 1 and in
1-81 Figures 6 and 7.

7. „In lines 575, 586, and perhaps other places in the manuscript, please consider that to most non-specialist readers, an 'ice layer' is a solid piece of plane ice, as on a frozen puddle. Perhaps another term, such as 'layer of ice particles´ or similar would be clearer to the non-specialists?"

   We agree and have chosen the terminology 'layer of ice particles´.

8. „At the end of the Introduction, in line 571, or in the Outlook and Conclusion, consider adding the approximate price per unit and the approximate operation costs. I would, however, understand why these numbers might be difficult or awkward to specify."

   We are reluctant of giving a price per unit since this may change substantially with time. For example, there are some activities to perhaps develop the laser for medical and/or space applications. This would presumably reduce the costs per laser unit drastically. Regarding operation cost, this concerns basically the costs for power consumption which is appr. 500 Watt under full operation (see legend to Figure 1).

9. We have corrected the typos as noted by the reviewer.

---

## Author Comment (AC2)

2-0 # Reply to reviewer's comments on the paper

2-1 ## 'VAHCOLI, a new concept for lidars: technical setup, science
2-2 applications, and first measurements'

2-3 submitted to AMT by Franz-Josef Lübken and Josef Höffner

2-4 Manuscript Number: amt-2021-33

2-5 ## Reviewer #2:

2-6

2-7 We appreciate the encouraging and positive comments from the reviewer. In particular we
2-8 are pleased that the reviewer has carefully read our manuscript and considers VAHCOLI
2-9 a '... revolutionary lidar instrument'. Furthermore, we thank the reviewer for his acknowl-
2-10 edgement of the fact that '... some basic knowledge has been recapitulated to guide the
2-11 reader.'

---

## Author Comment (AC3)

**Reply to reviewer's comments on the paper**

**'VAHCOLI, a new concept for lidars: technical setup, science applications, and first measurements'**

submitted to AMT by Franz-Josef Lübken and Josef Höffner

Manuscript Number: amt-2021-33

**Reviewer #3:**

We appreciate the encouraging and positive comments from the reviewer. We have taken the suggestions for improvements into account when preparing the revised version of the manuscript. We have marked the changes and respond to the reviewer's comments point by point in the following. In this response we repeat the reviewer's comments in blue, put our comments in green, and mark the modified version of the text in the manuscript in red. We have polished the text somewhat with the help of a native english speaker (shown in cyan in the manuscript).

*„The paper presents the development and first implementation of a lidar system that has the potential to transform observations of the middle atmosphere. The development of these compact and robust wind solid-state wind-temperature capable lidars is a significant technical achievement in itself. Combining precision resonance lidar techniques with Rayleigh and Mie techniques to yield winds and temperature in the troposphere, stratosphere, mesosphere, and lower thermosphere. The fact that these instruments are field deployable as compact units (1 m) allows deployments of distributed arrays of profilers that will support a variety of new investigations of the middle atmosphere. The paper provides a technical description of the instrument as well as a discussion of investigations that will supported (e.g. waves versus large scale turbulence). "*

*Specific Technical Comments*

1)+2) „The paper would benefit if some of the technical details were explained in greater detail, particularly in terms of the determination of the frequency stability of the seed laser and the confocal etalon using the Doppler-free spectroscopy (Figure 5). Does the locking use the Pound-Drever-Hall technique combined with the Doppler-free spectroscopy? What are the fundamental limitation of the tuning accuracy and precision based on the Doppler-free spectroscopy. How was the stability of the confocal resonator determined relative to that of the seed laser? The diagram of the lidar system could be presented in more detail as the schematic of record.

We understand the desire to learn more about instrumental details, in particular since several aspects are new to the lidar community. We note, however, that basically all components of VAHCOLI could be described in much more detail. This goes way beyond the scope of this paper and would, as we think, distract from the key point of this paper, namely to describe the principles of the technical subsystems and to give an overview of VAHCOLI and its atmospheric applications. Some

2-39 subsystems have already been described in more detail in the literature (see, for ex-
2-40 ample, references in the manuscript regarding the power laser) or will be published
2-41 in the near term future. Following the request of the reviewer, we have expanded the
2-42 description of the laser frequency control in this paper since this is fundamentally
2-43 different from most lidars and is a key feature of VAHOCLI. The main point is that
2-44 we do not stabilize the seeder laser (nor the power laser) to a single frequency (as
2-45 is done in several other lidars) but instead use Doppler-free spectroscopy (DFS) to
2-46 calibrate the frequency characteristics of the seeder laser (and thereby the power
2-47 laser) when its frequency is tuned up and down. We hope that the added text (see
2-48 below) helps to better understand the principle of frequency control of VAHCOLI.

2-49 „The individual peaks of the Doppler free spectrum serve as an absolute frequency
2-50 calibration for the seeder laser which is tuned up and down in frequency and fed
2-51 into the DFS system. Thereby the seeder laser frequency is known precisely (within
2-52 a few kHz) as a function of time and is subsequently used to control the frequency
2-53 of the power laser. The seeder laser also serves as a reference to lock a transmission
2-54 peak of the confocal etalon. Note that this procedure implies that (different to other
2-55 lidar systems) we do not lock the frequency of the seeder laser (nor the power laser)
2-56 to a single frequency."

2-57 3) „There are places where the the writing could be polished and made more concise."
2-58 We have improved the text with the help of a native english speaking colleague.

2-59

2-60 *Minor point*

2-61 „The authors note that the lasers were trucked from Aachen to Kühlungsborn ( 600
2-62 km) without significant misalignment. Can they determine the relative contribution
2-63 of the laser design, the driving skills of the shippers, and the quality of the autobahn
2-64 conditions to this result?"

2-65 Unfortunately, it is not possible to address the various contributions to the magni-
2-66 tude and the cause of the vibrations. We did not make any attempt to measure or
2-67 record the accelerations imposed to VAHOCLI during transport. However, we thank
2-68 the reviewer for this comment and will make an attempt to quantitatively record
2-69 the vibrations during the next shipment.